# Regional BOLD variability reflects microstructural maturation and neuronal ensheathment in the preterm infant cortex

Joana Sa de Almeida [1,2,3,11] ✉, Andrew Boehringer[2,11], Serafeim Loukas[2], Elda Fischi-Gomez [4,5,6], Annemijn Van Der Veek [2], Lara Lordier[1], Sebastien Courvoisier[7,8], François Lazeyras[7,8], Dimitri Van De Ville [7,8,9], Gareth Ball [3,10] & Petra S. Hüppi [1,2]

Blood Oxygen Level Dependent (BOLD) variability reflects meaningful brain activity, yet its structural and biological correlates during early development remain unknown. Using longitudinal resting-state fMRI and multi-shell diffusion imaging acquired longitudinally in 54 very preterm infants (at 33-weeks' gestational age and term-equivalent-age) and 24 full-term newborns, we investigated how BOLD variability evolves in very preterm infants, its relationship with cortical microstructure and gene expression, using the Brain-Span dataset, and how it differs from full-term newborns at term-equivalent age. During preterm development, BOLD variability increased in primary sensory-sensorimotor and proto-Default-Mode-Network regions, accompanied by decreases in cortical diffusivity. Gene expression analysis revealed concurrent upregulation of genes mediating gliogenesis and neuronal ensheathment. At term-equivalent age, very preterm infants showed decreased BOLD variability and increased cortical diffusivity, compared to full-term newborns. In this work, we show that BOLD variability reflects cortical microstructural maturation, mediated by upregulation of gliogenesis and neuronal ensheathment. Interruption of these processes by preterm birth identifies putative mechanisms of preterm brain injury.

Neural dynamics can be measured in a noninvasive, though indirect way, using functional MRI (fMRI) through the blood oxygen level dependent (BOLD) signal, measured with fMRI. Resting-state fMRI (RS-fMRI) has been a particularly attractive paradigm to obtain richly structured spontaneous activity. So-called "static" analysis techniques obtain measures that summarize the average behavior over a whole run.

However, the "BOLD signal variability," characterized by its variance, captures the magnitude of moment-to-moment regional variations in BOLD signal[1–3]. It is thought to indirectly reflect spontaneous

[1]Division of Development and Growth, Department of Women's, Children's, and Adolescent Health, University Hospitals of Geneva, Geneva, Switzerland. [2]Department of Pediatrics, Gynecology and Obstetrics, University of Geneva, Geneva, Switzerland. [3]Developmental Imaging, Murdoch Children's Research Institute, Melbourne, VIC, Australia. [4]CIBM Center for Biomedical Imaging, University of Lausanne, Lausanne, Switzerland. [5]Department of Radiology, University Hospital and University of Lausanne, Lausanne, Switzerland. [6]Signal Processing Laboratory, Ecole Polytechnique Fédérale de Lausanne (EPFL), Lausanne, Switzerland. [7]CIBM Center for Biomedical Imaging, University of Geneva, Geneva, Switzerland. [8]Department of Radiology and Medical Informatics, University of Geneva, Geneva, Switzerland. [9]Neuro-X Institute, Ecole Polytechnique Fédérale de Lausanne (EPFL), Geneva, Switzerland. [10]Department of Paediatrics, University of Melbourne, Melbourne, VIC, Australia. [11]These authors contributed equally: Joana Sa de Almeida, Andrew Boehringer. ✉e-mail: joana.alvessadealmeida@unige.ch

brain activity arising from moment-to-moment fluctuations in neuronal activity. In fact, BOLD variability has been shown to correlate with electrophysiological measures of neural dynamics, including EEG power and temporal signal variability[4–8]. These temporal fluctuations in neuronal activity, as measured by electrophysiological and EEG studies, are thought to contribute to synaptic connectivity and associate with better cognitive performance[4–6,9,10], thereby underlying brain function.

Although BOLD variability is affected by vascular factors, such as baseline cerebral blood flow and neurovascular coupling[11], studies combining BOLD and arterial spin labeling imaging show that group differences remain even after accounting for these vascular influences[12,13]. This evidence indicates that BOLD variability reflects not only vascular effects but also neural dynamics, supporting its use as a meaningful index of underlying functional brain activity.

In adults, BOLD variability varies across cortical regions, with greater variability in association and transmodal regions[8]. Greater BOLD variability has been linked to increased network organization, greater cognitive performance and increased task difficulty across different studies[2,3,8,13–18], highlighting its functional relevance.

BOLD variability typically increases during childhood[19] and serves as a strong predictor of age across the lifespan[1], then declining with aging[2,16,20,21]. To date, no studies have examined changes in BOLD variability in infants prior to term-equivalent age (TEA).

Additionally, few studies have explored the structural and biological underpinnings of BOLD variability, although evidence suggests it correlates with contiguous white matter (WM) microstructural integrity, both in adults and children[19,22,23]. Recently, using the ex vivo Big-Brain histological dataset[24], Baracchini et al. found that BOLD variability was increased in cortical areas with greater laminar differentiation and higher neuronal density, particularly in granular cortical regions where the distinct layer IV supports efficient sensory processing and a broader range of functional responses[16]. However, to date, no studies have examined the relationship between BOLD variability and intra-cortical microstructure, nor its biological correlates, specifically the underlying gene expression patterns, in adults or infants.

Diffusion-weighted imaging measures water diffusion in brain tissue at the mesoscopic scale[25], enabling the study of brain microstructure. However, investigating gray matter (GM) microstructure with diffusion MRI (dMRI) poses significant challenges. Unlike WM, GM does not display orientational coherence, is not as heavily myelinated, and is characterized by a complex intra- and extracellular environment, with multiple barriers and different cellular compartments.

Traditional signal representation models of diffusion, such as diffusion tensor imaging (DTI), provide insights into GM organization by measuring the anisotropic diffusion of water molecules within the tissue, without intrinsic diffusivity assumptions, capturing variations in diffusion patterns[26]. However, DTI assumes a Gaussian distribution of diffusion, which does not hold true in many biological tissues. To quantify the non-Gaussianity of diffusion, diffusion kurtosis imaging (DKI) has been proposed as an extension of DTI, accounting for the non-Gaussian signal decay and estimating diffusion kurtosis as a reflective marker for tissue heterogeneity[27]. Studies have shown that diffusion kurtosis is more sensitive to microstructural complexity within tissues than DTI[28,29]. Both DTI and DKI lack specificity for multiple diffusion signals within a single voxel (e.g., crossing fibers) and complex tissue microstructural features. To overcome this, multi-compartment models have been proposed to provide a more complete characterization of tissue complexity. The spherical mean technique (SMT) model estimates microstructural parameters from the direction-averaged diffusion signal, hence removing directionality dependence, and provides measures sensitive to cellular composition, including intra-axonal and extra-axonal compartments[30]. Unlike other multi-compartment models, such as NODDI (neurite orientation dispersion and density imaging), SMT does not rely on strong assumptions regarding fixed diffusivities, making it more suitable for complex tissue environments, like the GM[31]. Taken together, the combination of different dMRI models might provide valuable insights into GM microstructure and its association with BOLD variability.

While imaging can reveal the relationship between BOLD variability and cortical microstructure maturation during early brain development, gene expression analysis can provide mechanistic insight into the processes and pathways underlying developmental functional and structural changes occurring in the cortex. We hypothesized that regional changes in BOLD variability, coupled with changes in the cortical microstructure, are associated with the expression of genes linked to neocortical organization.

In addition to the lack of evidence about how BOLD variability changes during early brain development, few studies have examined the impact of preterm birth on its developmental trajectory[24,32]. Preterm birth is known to occur during a critical period of activity-dependent plasticity and brain development, characterized by axonal and dendritic growth and arborization, synaptogenesis, neural organization and myelination[33,34]. Preterm birth exposes the infants to a dramatic environmental change and leads to altered structural and functional brain development with adverse, long-term outcomes[35–43].

In this study, we aimed to investigate the maturation of BOLD variability in very preterm (VPT) infants, longitudinally, from 33 to 40-weeks gestational age (GA), across different resting-state networks (RSNs). In addition, using a comprehensive dMRI analysis, combining metrics from DTI, DKI, and SMT models, we evaluated how observed changes in BOLD variability across the cortex align with regional measures of cortical microstructure. Furthermore, to better understand the biological correlates of the observed functional and microstructural changes, we examined spatiotemporal patterns of gene expression in postmortem tissue samples over the same time period. Finally, we aimed to assess whether preterm birth impacts the expected developmental maturation of BOLD variability and cortical microstructure, in comparison to full-term (FT) birth.

## Results
### Patient demographics
In both RS-fMRI and dMRI samples, significant differences between the VPT and FT groups were observed, as expected, in the following perinatal variables: GA at birth, birth weight, birth height, birth head circumference, APGAR at 1 and 5 min and incidence of bronchopulmonary dysplasia (BPD). There were no differences between groups in sex, GA at 2nd MRI scan (TEA), neonatal asphyxia, intrauterine growth restriction, intraventricular hemorrhage grade I, and socio-economic parental status score[44], (Table 1 and Supplementary Table S1). There were no significant differences, for any perinatal clinical variable, between the RS-fMRI and dMRI samples, in the VPT infants' group or the FT newborns' group (Table 1 and Supplementary Table S2).

### Longitudinal cortical BOLD variability and diffusion microstructural changes during early preterm brain development
In the VPT cohort, BOLD variability (estimated as BOLD SD) increased significantly (false discovery rate (FDR)-corrected; $p < 0.05$) from 33- to 40-weeks GA (wGA) in the primary sensory (visual, auditory), sensorimotor and proto-default-mode-network (pDMN) (composed of the posterior cingulate cortex and precuneus) (group 1). In contrast, BOLD variability did not change significantly in paralimbic, thalamus, limbic or prefrontal cortex networks (group 2), (Fig. 1 and Supplementary Table S3).

Complementary analysis using ALFF and fALFF to estimate BOLD variability yielded consistent longitudinal patterns across the same regions (see Supplementary Fig. S1).

We next examined changes in diffusion microstructural metrics in VPT infants, from 33- to 40-wGA, across all cortical regions. Significant

**Table 1 | Clinical perinatal characteristics of (a) RS-fMRI and (b) dMRI final samples, for both VPT infants and FT newborns**

| | | Very Preterm (VPT) | Full-term (FT) | *p*-value[c] | *p*-value[d] | *p*-value[e] |
|---|---|---|---|---|---|---|
| Clinical characteristics | a) | RS-fMRI, *n* = 31 | a) RS-fMRI, *n* = 19 | VPT vs. FT | VPT group RS-fMRI vs. dMRI | FT group RS-fMRI vs. dMRI |
| | b) | dMRI, *n* = 39 | b) dMRI, *n* = 18 | | | |
| GA at birth | a) | 29.28 ($\pm$1.99) | 39.42 ($\pm$1.10) | 0.001* | 0.824 | 0.763 |
| Weeks, mean (SD) | b) | 29.17 ($\pm$2.16) | 39.53 ($\pm$1.13) | 0.001* | | |
| GA at birth | a) | $24^{4/7}$–$32^{4/7}$ | $37^{1/7}$–$41^{1/7}$ | | | |
| Weeks, range | b) | $24^{1/7}$–$32^{4/7}$ | $37^{1/7}$–$41^{1/7}$ | | | |
| GA-33w MRI scan | a) | 33.63 ($\pm$0.44) | | | 0.582 | |
| Weeks, mean (SD) | b) | 33.57 ($\pm$0.45) | | | | |
| GA-33w MRI scan | a) | $32^{6/7}$–$34^{3/7}$ | | | | |
| Weeks, range | b) | $32^{4/7}$–$34^{3/7}$ | | | | |
| GA-TEA MRI scan | a) | 40.16 ($\pm$0.52) | 40.08 ($\pm$0.77) | 0.692 | 0.907 | 0.695 |
| Weeks, mean (SD) | b) | 40.17 ($\pm$0.56) | 40.18 ($\pm$0.76) | 0.949 | | |
| GA-TEA MRI scan | a) | $38^{5/7}$–$41^{0/7}$ | $39^{0/7}$–$41^{3/7}$ | | | |
| Weeks, range | b) | $38^{5/7}$–$41^{1/7}$ | $39^{0/7}$–$41^{3/7}$ | | | |
| Sex: female/male | a) | 14(45)/17(55) | 13(68)/6(32) | 0.19 | 0.872 | 0.661 |
| *n* (%) | b) | 20(51)/19(49) | 10(56)/8(44) | 0.98 | | |
| SES | a) | 4.87 ($\pm$3.40) | 3.47 ($\pm$2.80) | 0.141 | 0.99 | 0.948 |
| Mean (SD) | b) | 4.74 ($\pm$3.15) | 3.39 ($\pm$2.93) | 0.129 | | |
| Birth weight | a) | 1231 ($\pm$372) | 3226 ($\pm$364) | 0.001* | 0.869 | 0.661 |
| Grams, mean (SD) | b) | 1216 ($\pm$388) | 3282 ($\pm$404) | 0.001* | | |
| Birth height | a) | 38.29 ($\pm$3.45) | 49.87 ($\pm$2.1) | 0.001* | 0.690 | 0.812 |
| Centimeter, mean (SD) | b) | 37.91 ($\pm$4.29) | 50.03 ($\pm$1.99) | 0.001* | | |
| Birth head circumference | a) | 26.46 ($\pm$2.31) | 34.58 ($\pm$1.15) | 0.001* | 0.634 | 0.545 |
| Centimeter, mean (SD) | b) | 26.76 ($\pm$2.79) | 34.83 ($\pm$1.38) | 0.001* | | |
| APGAR score 1 min | a) | 5.48 ($\pm$3.19) | 8.74 ($\pm$1.56) | 0.001* | 0.843 | 0.809 |
| Mean (SD) | b) | 5.33 ($\pm$3.12) | 8.61 ($\pm$1.58) | 0.001* | | |
| APGAR score 5 min | a) | 8.16 ($\pm$1.75) | 9.58 ($\pm$0.84) | 0.002* | 0.357 | 0.779 |
| Mean (SD) | b) | 7.72 ($\pm$2.15) | 9.50 ($\pm$0.86) | 0.001* | | |
| IUGR | a) | 5 (16) | 0 | 0.174 | 1 | |
| *n* (%) | b) | 6 (15) | 0 | 0.195 | | |
| Neonatal asphyxia | a) | 0 | 0 | | | |
| *n* (%) | b) | 0 | 0 | | | |
| BPD | a) | 12 (38.7) | 0 | 0.006* | 1 | |
| *n* (%) | b) | 15 (38.5) | 0 | 0.006* | | |
| IVH (grade 1) | a) | 3 (9.7) | 0 | 0.432 | 1 | |
| *n* (%) | b) | 3 (7.7) | 0 | 0.568 | | |

Group-characteristics per MRI sequence sample [(c) VPT vs. FT, within RS-fMRI and dMRI samples], as well as differences between MRI sequence samples [RS-fMRI vs. dMRI within each group: (d) VPT and (e) FT], were compared using two-sided independent samples t-test for continuous variables and chi-squared test for categorical variables. Significant differences are indicated with (*), *p* < 0.05. Source data are provided as a Source data file.

increases in BOLD variability were associated with an overall decrease in cortical diffusivity and kurtosis metrics (group 1, Fig. 2).

In group 1, where BOLD variability increased significantly, SMT metrics globally decreased, namely intrinsic diffusivity (diff), extra-neurite mean diffusivity (extraMD), extra-neurite transverse diffusivity (extraTrans) and intra-neurite volume fraction (intra). Additionally, DTI showed significant decreases of both fractional anisotropy (FA) and mean diffusivity (MD) in all these regions. DKI revealed significant decreases in radial kurtosis (RK) and/or mean kurtosis (MK) across these group 1 regions.

In group 2, where BOLD variability did not change significantly, the microstructural changes were less coherent. More specifically, the thalamus displayed a significant increase in FA, accompanied by significant decreases in diff, extraMD, extraTrans and MD between 33- and 40-wGA. In contrast, the PFC displayed a marked decrease in both FA and intra-neurite volume fraction, accompanied by an increase in MD. Both limbic and paralimbic regions displayed an overall decrease

in SMT diffusivities with decreased intra-neurite volume fraction and FA (Fig. 2).

We conducted an additional analysis, clustering cortical regions based on the similarity of their longitudinal microstructural maturation, rather than BOLD variability, to compare the microstructural clusters with those derived from BOLD changes. We found that the microstructural patterns are consistent with the functional partition (Supplementary Fig. S3).

### Relationship between longitudinal cortical microstructural changes and BOLD variability delta changes

To identify the microstructural metrics that best predicted the observed changes in BOLD variability, we employed lasso-based sparse multivariate regression models on the regional-average data. Out of the 9 microstructural metrics, 4 features (MD, RK, AK, and FA) were retained, at the optimal pure LASSO penalty ($\lambda_1 se = 0.015$, $\alpha = 1$), as predictors of regional BOLD variability changes from 33- to 40-wGA.

**Fig. 1 | Longitudinal regional BOLD variability changes in VPT infants from 33 weeks GA to TEA. a** Bar plots illustrate the group-averaged BOLD variability longitudinal changes in VPT infants ($n = 31$) from 33-wGA to TEA for each RSN, scaled to the largest absolute mean regional change. Embedded boxplots show the distribution of individual subject changes within each RSN, with individual values scaled to the largest absolute subject-level change observed across all RSNs for that metric. The center line of each boxplot represents the median (50th percentile), the box bounds indicate the 25th–75th percentiles, and the whiskers represent the minimum and maximum values within 1.5 × IQR. Non-scaled boxplots can be found in Supplementary Fig. S2. Regional measurements were derived from the same MRI scans for each participant. Statistical analysis: two-sided paired t-tests were performed comparing 33-wGA to TEA data within each RSN, and $p$-values were FDR-adjusted for multiple comparisons. Significant changes are indicated by asterisks ("*"$p < 0.05$, "**"$p < 0.01$, "***"$p < 0.001$, after FDR correction). Regions were grouped according to the presence (group 1) or absence (group 2) of significant delta changes in BOLD variability. **b** Brain plots illustrating the RSNs with significant longitudinal BOLD variability changes in VPT infants from 33-wGA to TEA. The networks include PCUN, PCC, SSM, VIS, and AUD, with color mapping representing the magnitude of change (increase in red, decrease in blue). VPT very preterm infants, PCUN precuneus, PCC posterior cingulate cortex, SSM sensorimotor, VIS visual, AUD auditory, THAL thalamus, PFC prefrontal cortex. Source data are provided as a Source data file.

MD showed the strongest association ($\beta = 0.419$), followed by RK ($\beta = 0.402$), AK ($\beta = 0.353$), and FA ($\beta = 0.332$). The model achieved a high training fit ($R^2 = 0.965$); however, leave-one-region-out cross-validation indicated that these longitudinal changes explained 28% of the observed variance in BOLD variability ($R^2_{loo}c_v = 0.28$, RMSE = 0.054). Using less penalized models—a lasso elastic net ($\alpha = 0.5$, $\lambda\_min = 0.0032$) and a—additional metrics (intra, diff, extraMD, and MK) were retained, increasing the explained variance to 61–64% ($R^2_{loo}c_v = 0.61$–0.64, RMSE = 0.0295–0.032, respectively).

**Genetic expression patterns during early brain development**

Using a developmental transcriptomic dataset of bulk tissue mRNA data sampled from cortical tissue in 18 post-mortem prenatal human specimens aged 8–37 post conceptional weeks (pcw) (http://development.psychencode.org/), we compared genetic expression patterns between group 1 (regions where BOLD variability increased significantly from 33- to 40-wGA) and group 2 (regions with no significant longitudinal BOLD variability changes), from mid- to late-fetal period. We hypothesized that regions with changes in BOLD variability (group 1), which also presented coupled cortical microstructural maturational changes, would display an increase in expression of genes linked to neocortical organization.

From mid- to late-fetal period (16–22 to 35–37 pcw), 786 out of the total 5287 genes displayed a significant group*time interaction, after correction for multiple comparisons (FDR-corrected; $p < 0.05$). From these genes, using pairwise comparisons to filter those with differential expression between group 1 and 2 from mid- to late-fetal, we identified 132 genes that increased significantly more in group 1 vs. 2, whereas only 9 genes increased significantly more in group 2 vs. group 1. Additionally, 64 genes were found to decrease significantly more in group 1 vs. 2, and 45 genes decreased significantly more in group 2 vs. 1.

Using geneset enrichment analysis[45], with a background set of 5287 genes[46], we identified significant enrichments for several important neocortical organizational processes in genes that increased significantly more in group 1 than 2. These included: "neuronal ensheathment" (FDR = 0.000014, enrichment = 6.15, 14 genes), "gliogenesis" (FDR = 0.018, enrichment = 2.85, 14 genes, from which 7 genes are common to neuronal ensheathment), "extracellular structure organization/external encapsulating structure organization" (FDR = 0.018, enrichment = 3.58, 10 genes), "cytokine-mediated signaling pathway" (FDR = 0.008, enrichment = 3.6, 13 genes), "coagulation/wound healing" (FDR = 0.008/0.01, enrichment = 4.09/3.03, 10/14 genes, where the 10 genes from coagulation are common to both categories), "extrinsic apoptotic signaling pathway" (FDR = 0.01, enrichment = 3.8, 10 genes) and "muscle cell proliferation (FDR = 0.02, enrichment = 3.8, 9 genes) (Fig. 3c). A list of all significant categories and their assigned genes can be found in Supplementary Table S4. No significant enrichment analysis results were found for the genes that increased significantly more in group 2 than 1, or for the genes that decreased significantly differently between group 1 and 2.

The combined list of enriched genes associated with neuronal ensheathment and gliogenesis categories comprised a total of 27 genes, including constituents of the myelin sheath (*MBP, MAG, MAL, MOBP, PLLP, PLP1, PMP22,* and *CLDN11*), genes with a role in myelination regulation (*NKX6-2, UGT8,* and *GPR17*), important for oligodendrocyte function and expression (*MBNL2, APOD,* and *VWA1*), for tissue integrity (*COL6A1*), with a role in growth factor signaling (*ERBB3*), neuroprotection and/or glioprotection (*TNFRSF1B, GPR37L1,* and *IL6ST*), neuronal and glial differentiation and development (*GPR183, APCDD1, NDRG2, CSRP1, TSC22D4,* and *CD9*) and in fatty acid and sphingolipid biosynthesis (*ELOVL1, PADI2*). ANOVA followed by Tukey's test identified a significant increase in expression of this group of genes in both groups (1 and 2), specifically from mid- to late-fetal (T1 to T2, group 1: z-ratio = −9.84, $p = 5.88 \times 10^{-14}$, group 2: z-ratio = −7.15, $p = 1.27 \times 10^{-11}$). No significant changes were observed from early- to mid-fetal in both groups (T0 to T1, z-ratio = −0.59 to −0.33, $p > 0.05$). The increase from T1 to T2 was significantly higher in group 1, compared to group 2 (z-ratio = −9.77, $p < 1.5 \times 10^{-22}$). At the late-fetal timepoint (T2), these same genes were also expressed at significantly higher levels in group 1 compared to group 2 (z-ratio = 7.74, $p = 1.80 \times 10^{-13}$). In contrast, there were no differences in the

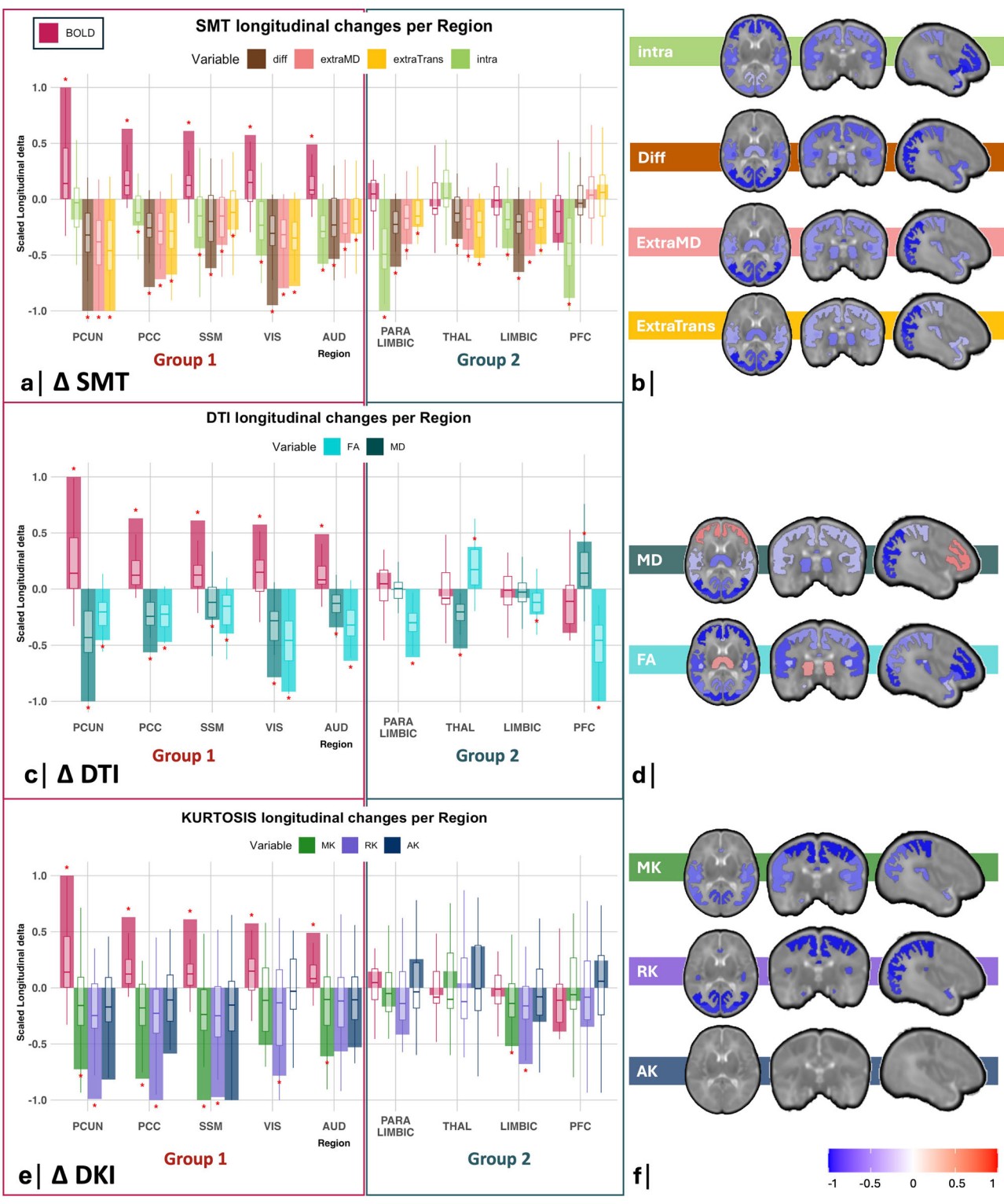

**a | Δ SMT** — SMT longitudinal changes per Region

**b |**

**c | Δ DTI** — DTI longitudinal changes per Region

**d |**

**e | Δ DKI** — KURTOSIS longitudinal changes per Region

**f |**

expression of these genes between groups at T0 (z-ratio = −2.7, $p = 0.073$) and at T1 (z-ratio = −2.23, $p = 0.22$), (Fig. 3d). The expression of each of these genes at each time-point and per group is shown in Fig. 3e.

### Effect of preterm birth on BOLD variability and cortical microstructure at TEA

Compared to FT newborns, VPT at TEA showed decreased BOLD variability in the thalamus, PCC, visual, auditory and limbic networks

(Fig. 4a). Complementary analysis using ALFF and fALFF to estimate BOLD variability yielded similar cross-sectional patterns across the same regions (see Supplementary Fig. S4). Regarding cortical microstructural differences, in comparison to FT, VPT at TEA displayed overall increased diffusivities (extra-neurite mean diffusivity, extra-neurite transverse diffusivity, intrinsic diffusivity, and MD) across all RSNs (Fig. 4b, c). VPT at TEA also presented increased kurtosis (RK and/or MK) across RSNs cortical regions, apart from the thalamus (Fig. 4d). There were no significant differences between groups

**Fig. 2 | Longitudinal regional cortical microstructural diffusivity changes in VPT infants from 33 weeks GA to TEA.** Bar plots illustrate the group-averaged BOLD variability ($n = 31$) and microstructural diffusivities ($n = 39$) longitudinal changes in VPT infants from 33-wGA to TEA across RSNs, with each metric scaled to its own largest absolute mean regional change. Embedded boxplots show the distribution of individual subject changes within each RSN for each metric, with individual values scaled to the largest absolute subject-level change observed across all RSNs for that metric. The center line of each boxplot represents the median (50th percentile), the box bounds indicate the 25th–75th percentiles, and the whiskers represent the minimum and maximum values within 1.5 × IQR. Regional measurements were derived from the same MRI scans for each participant. **a** SMT metrics include intra-neurite volume fraction (intra), intrinsic diffusivity (diff), extra-neurite mean diffusivity (extraMD) and extra-neurite transverse diffusivity (extraTrans), **c** DTI metrics include mean diffusivity (MD) and fractional anisotropy (FA); **e** Microstructural DKI metrics include mean (MK), radial (RK) and axial kurtosis (AK). Non-scaled boxplots can be found in Supplementary Fig. S2. **b**, **d**, **f** Brain plots illustrating the RSNs with significant cortical microstructural changes in VPT infants from 33 -GA to TEA. Statistical analysis: two-sided paired t-tests were performed comparing 33-wGA to TEA within each RSN, and p-values were FDR-adjusted for multiple comparisons. Significant changes are highlighted by a "*" ($p < 0.05$, after FDR correction). Regions were grouped according to the presence (group 1) or absence (group 2) of significant delta changes in BOLD variability. VPT very preterm infants, TEA term-equivalent age, PCUN precuneus, PCC posterior cingulate cortex, SSM sensorimotor, VIS visual, AUD auditory, THAL thalamus, PFC prefrontal cortex. Source data are provided as a Source data file.

regarding AK across all cortical regions. Details of all regional differences per diffusion metric, between VPT and FT infants, are provided in Supplementary Table S5.

## Discussion

In this study, we combined in vivo fMRI and dMRI data with independent ex vivo gene expression analyses to explore the regional maturation of BOLD variability across RSNs in very preterm infants, longitudinally, from 33 wGA to TEA. Our findings reveal an alignment between brain developmental changes in BOLD variability and regional cortical microstructural maturation, as assessed through advanced dMRI models, including DTI, DKI, and SMT. Furthermore, we identified specific spatiotemporal patterns of gene expression that may underlie the observed changes.

Additionally, using a cross-sectional approach at TEA, we demonstrate that preterm birth is associated with changes in the typical maturation pattern of BOLD variability and cortical microstructure, in comparison to FT birth.

### Cortical microstructural maturation underlies RSNs BOLD variability increases during preterm infants' brain development

From 33-wGA to TEA, BOLD variability significantly increased in preterm infants, specifically in the primary sensory (visual and auditory), sensorimotor and proto-DMN (precuneus and PCC) (group 1). The primary visual, auditory and somatosensory regions are known to present an early establishment of activity-dependent thalamo-cortical connectivity. In fact, from 24- to 32-pcw, thalamocortical fibers grow into and accumulate in the subplate, beginning in somatosensory areas[47], followed by the visual[48] and auditory[49] cortices, and leading to the formation of synapses in the deep cortical plate during prenatal development[50]. During this period, electrophysiological phenomena can be detected by EEG, with the appearance of evoked potentials and giant transients, indicating synaptogenesis, cortical activity and processing of sensory information[51,52]. Indeed, during brain development, synaptogenesis was shown to follow a spatiotemporal pattern, occurring earlier in primary motor and sensory areas, and later in the prefrontal cortex[53]. Our findings, revealing significant longitudinal increases in BOLD variability specifically in primary motor, sensory, and posterior brain regions from 33 wGA to TEA, with no changes in prefrontal regions, agree with the spatiotemporal pattern of synaptogenesis and cortical activity. These results, quantifying fluctuations in BOLD signal amplitude, complement previous literature findings showing that the temporal complexity of BOLD signals, evaluated by means of the Hurst Exponential, also increases overall in very preterm infants' GM from birth until TEA[54,55]. Similar to our findings, the greatest increases in Hurst Exponential were also found in motor and visual networks, compared to frontal networks, suggesting that primary sensorimotor areas mature from "more random" to "more ordered" states faster than the frontal ones[54]. BOLD variability provides thus a complementary perspective, suggesting that primary sensorimotor regions also present increasingly variable functional activity, highlighting their accelerated functional maturation.

Regional functional BOLD variability increases in group 1 were accompanied by significant decreases in cortical microstructural diffusivity and kurtosis, as well as by decreases in FA and intra-neurite volume fraction, evaluated using different dMRI models (DTI, DKI, and SMT).

Decreases in cortical diffusivity (MD, intrinsic diffusivity, and extra-neurite mean diffusivity) from 33-wGA to TEA align with an ongoing microstructural cortical maturation, possibly relating to an increased dendritic arborization, formation of basal dendrites cross-connections and gliogenesis that occur during this period[56–59]. Previous studies support the notion that MD decreases in the human cortex throughout preterm development[60]. Decreases in cortical MD have been linked to an increase in dendritic density and synaptic complexity, as observed in histological studies of rodents[61]. In addition, we have previously shown that these regions in group 1, namely the visual, auditory, sensorimotor, precuneus and cingulate cortices, undergo a significant increase in diffusion ODI (orientation dispersion index) and FC (fiber cross-section) from 33-wGA to TEA, further supporting an increase in cortical complexity and cellular density during this period[42].

The observed decreases in cortical MK are consistent with previous findings evaluating human brain prenatal development[62]. This is in contrast with the known increase in MK in the cortex observed later, after birth, during the postnatal period and childhood, especially during the first 2 years of life[29,63,64]. The observed decrease in MK during the prenatal period was hypothesized to indicate a continuous decrease in diffusion barriers, possibly caused by a decrease in neuronal density during this period[65–67]. In agreement with this theory is the observed longitudinal decrease of intra-neurite volume fraction across these cortical regions, which is in line with previous studies showing a decrease in neurite density index (NDI) across GM tissue during this period[68]. Decreases in cortical neuronal density could be explained by synaptic pruning and physiological apoptosis, involution of the subplate, as well as by a cortical expansion, which may lead to an apparent decrease in neuronal density[69]. Additionally, decreases in intra-neurite volume fraction may be led by the disruption of the radial glial scaffolding, known to occur during this period of development, and contribute to a reduction of cortical cell density[56,70]. In fact, it is unlikely that measures, such as intra-neurite volume fraction and NDI, solely reflect neurite structures, with possible contributions from prominent radial glial fibers, which exhibit elongated, coherent radial organization, and may be interpreted as intra-neurite compartments. Interestingly, we show that the decreases in MK are mostly led by decreases in RK, and not AK. These kurtosis reductions, particularly in the radial direction, support that disruption of the radial glia scaffolding may be at also the origin of the observed MK decrease.

The overall decreases in cortical diffusivity, kurtosis and in intra-neurite volume fraction were also accompanied by decreases in FA in

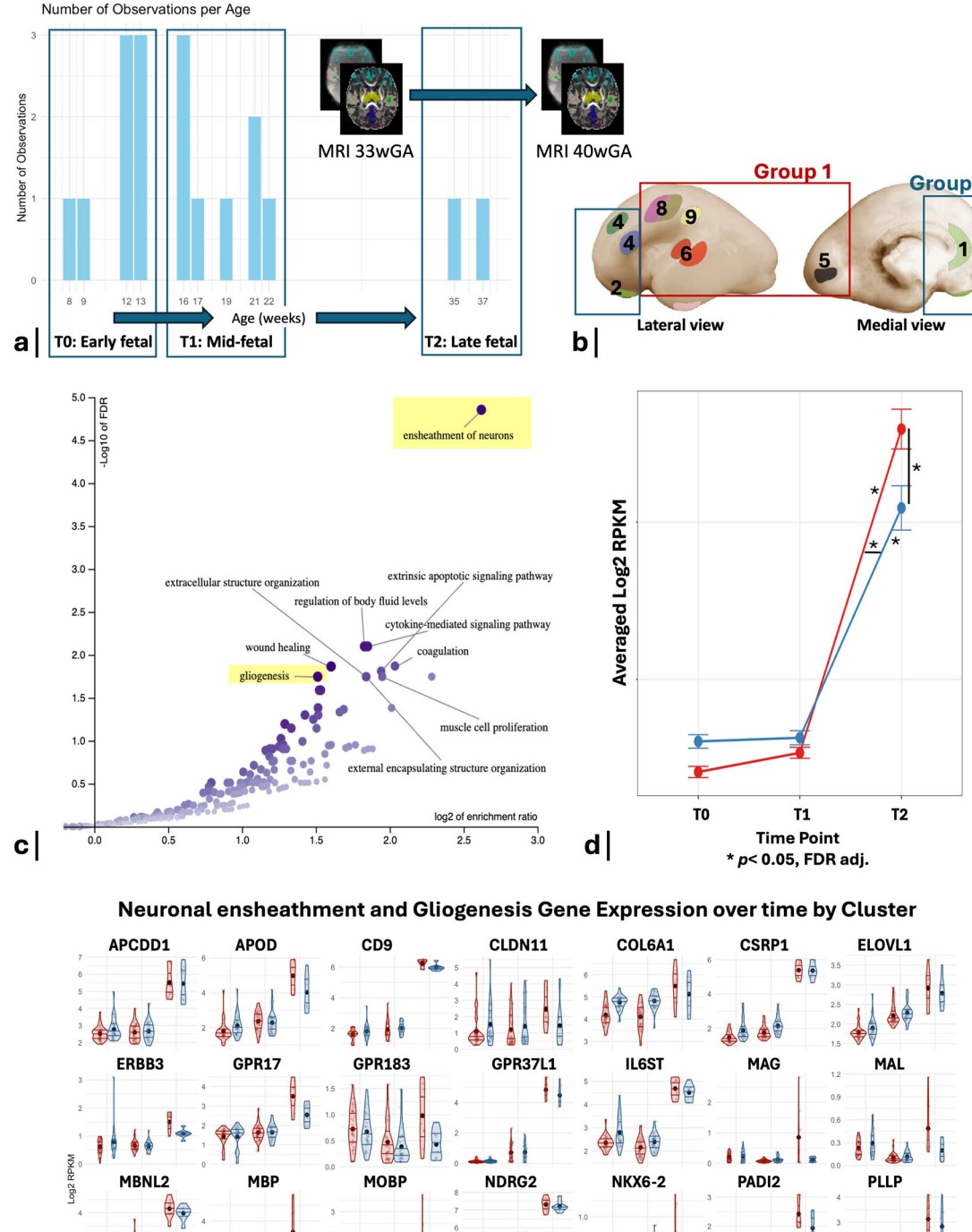

**Neuronal ensheathment and Gliogenesis Gene Expression over time by Cluster**

group 1 regions. The FA reduction is in alignment with previous studies showing decreased cortical anisotropy during prenatal development, attributed to reduced radial glia scaffolding and increasing dendritic complexity[60,61,70–72].

Group 2 cortical regions, where BOLD variability did not change significantly from 33-wGA to TEA, exhibited heterogenous diffusion changes, with overall less pronounced diffusivity decreases. In

particular, the PFC and the paralimbic region (combining the OFC and the temporal pole) showed marked FA and intra-neurite volume fraction reductions, exceeding diffusivity changes. Literature indicates that frontal association areas have lower dendritic density, synapse number and glia density than primary sensory regions in neonates[53,73,74]. This could explain the less pronounced decreases in diffusivity in frontal areas. Furthermore, the marked decline in FA and

**Fig. 3 | Regional genetic expression patterns during early brain development.**
**a** Bar plot showing the number of observations (specimen ID) per gestational age (in weeks), obtained from the BrainSpan developmental transcriptomic dataset (http://development.psychencode.org/). The transcriptomic data were grouped per period of gestation (T0: early-fetal, 8–13 pcw; T1: mid-fetal, 16–22 pcw; T2: late-fetal, 35–37 pcw). The longitudinal brain MRI scans were acquired from our preterm infants' cohort at a first time-point between the mid- and late-fetal periods (33th wGA) and a second one just after the late-fetal period (40th wGA). **b** Topographic locations of the samples from the BrainSpan developmental transcriptomic dataset. Schematic shows anatomical positions of dissections of the prenatal brain samples, which are coded according to their correspondence to our RSNs brain atlas: (1) medial frontal cortex (MFC) = limbic, (2) orbitofrontal cortex (OFC) = paralimbic belt, (4) dorsolateral and ventrolateral prefrontal cortex (DLPFC + VLPFC) = PFC, (5) primary visual cortex (V1) = visual, (6) primary auditory cortex (A1C) = auditory, (8) primary motor and primary somatosensory cortex (M1 + S1) = sensorimotor, (9) inferior parietal cortex (IPC) = precuneus. These regions

were grouped according to the presence (group 1) or absence (group 2) of significant BOLD variability changes from 33- to 40-wGA. Image adapted from ref. 123. **c** Volcano plot showing significant enrichment of gene ontology terms (biological processes) (FDR-corrected, $p < 0.05$) of the genes expressed significantly higher in group 1 than group 2 from mid- to late fetal period. Gene ontology enrichment analysis was performed using WebGestalt (Fisher's exact test, FDR < 0.05). **d** Line graph representing the combined expression (averaged Log2 RPKM) of the 27 genes belonging to neuronal ensheathment and/or gliogenesis categories per time-point (T0: early-fetal; T1: mid-fetal; T2: late-fetal) and per group (1 in red; and 2 in blue). Two-way ANOVA with post-hoc Tukey's test for pairwise comparison was performed. Mean and 95% confidence intervals are illustrated in red for group 1 and in blue for group 2. Significant results are highlighted by a "*" (FDR-corrected, $p < 0.05$). **e** Violin plots illustrating the expression (in Log2 RPKM) of each of the 27 genes belonging to neuronal ensheathment and/or gliogenesis categories across time points (T0, T1, and T2), per group (1: group 1; 2: group 2). Source data are provided as a Source data file.

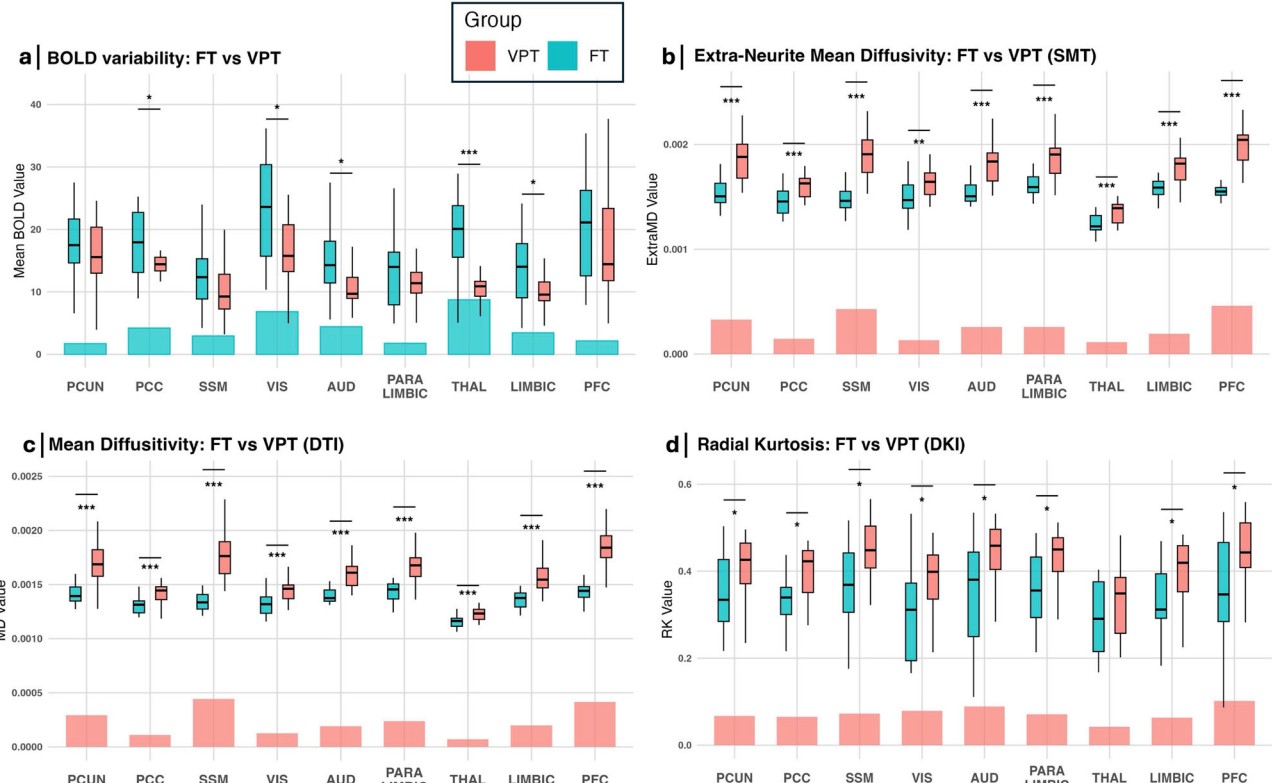

**Fig. 4 | Effect of preterm birth on BOLD variability (FT, $n = 19$; VPT, $n = 31$) and cortical microstructure (FT, $n = 18$; VPT, $n = 39$) in infants at TEA.** Boxplots represent the distribution of individual subject values for FT (in blue) and VPT at TEA infants (in coral) within each network, illustrating inter-subject variability. The center line of each boxplot represents the median (50th percentile), the box bounds indicate the 25th–75th percentiles, and the whiskers represent the minimum and maximum values within 1.5 × IQR. Bar plots illustrate the average group difference (FT-VPT) for each network. Regional measurements were derived from

the same MRI scans for each participant. **a** BOLD variability, **b** extra-neurite mean diffusivity, **c** mean diffusivity and **d** radial kurtosis. Statistical analysis: two-sided independent t-tests were performed comparing FT and VPT groups within each RSN, and $p$-values were FDR-adjusted for multiple comparisons. Lines indicate significant differences between groups with *$p < 0.05$, **$p < 0.01$, and ***$p < 0.001$, FDR adjusted. VPT very preterm infants, FT full-term infants, PCUN precuneus, PCC posterior cingulate cortex, SSM sensorimotor, VIS visual, AUD auditory, THAL thalamus, PFC prefrontal cortex. Source data are provided as a Source data file.

NDI in frontal regions has been consistently observed in other human prenatal studies, and has been shown to correlate with increased GM volume and mean curvature[60,68,75]. This suggests that cortical volume expansion would be driven by dendritic arborization and neuropil expansion, rather than the addition of new cells[76]. When volume expansion outpaces the cellular density increases, FA and intra-neurite volume fraction decline, while MD may increase, as observed in the PFC. These microstructural changes indicate overall delayed maturation in more frontal regions, potentially explaining the absence of significant BOLD variability increases in these regions.

Interestingly, the thalamus showed increased FA with decreased diffusivity. In fact, as a subcortical region, the thalamus exhibits a distinct microstructural organization in comparison to cortical GM areas, which can explain these findings. Despite the important establishment of thalamocortical connectivity during this period[77–79], no significant increase in BOLD variability was observed in this region.

To better understand which diffusion metrics most effectively explain the observed regional increases in BOLD variability, we employed lasso-based sparse multivariate regression models. Diffusion- and kurtosis-derived microstructural metrics emerged as strong

contributors. Under strict penalization, a limited set of predictors (MD, RK, AK, and FA) was retained, explaining up to 28% of the observed changes in BOLD variability. Relaxing model sparsity increased the number of contributing microstructural metrics, including intra, diff, extraMD, MK, and substantially improved predictive performance up to 61–64%, suggesting that BOLD variability reflects a distributed and correlated microstructural maturation process. Together, these findings support an important role for cortical microstructural maturation in shaping functional variability.

Decreases in diffusion-related metrics and, in particular, the strong association with the decrease in MD, suggest that an increase in dendritic arborization, dendritic density, synaptogenesis and gliogenesis[61] may contribute to BOLD variability increase. In fact, the intracortical burst of dendritic growth and maturation is known to occur in parallel with rapid synaptogenesis during the late mid-fetal and early late-fetal periods, coinciding with changes in functional properties of the neuronal cortical circuitry and appearance of evoked potentials[80,81]. Furthermore, the association of increased BOLD variability with decreases in kurtosis-related metrics and FA denote a decrease in diffusion barriers, possibly due to the disruption of the radial scaffold with the ongoing cortical maturation, allowing the extension of dendritic arborization, and its differentiation into more localized glia cells once cortical migration finishes, ultimately supporting synaptogenesis[82,83]. In fact, literature suggests that regions where radial glia disappear earlier tend to exhibit earlier synaptic activity[82,84,85].

### Genes mediating gliogenesis and neuronal ensheathment increase more in RSNs where BOLD variability increased

Between 16–22 pcw and 35–37 pcw, corresponding to the mid- to late-fetal period, analysis of ex vivo fetal cortical gene expression patterns revealed a significant upregulation of genes associated with neuronal ensheathment and gliogenesis. These findings are consistent with previous literature showing a marked increase in myelination-related gene expression starting from the mid-fetal period, continuing until birth, and into the post-natal period[86].

This increase in genes associated with neuronal ensheathment and gliogenesis, from mid- to late-fetal period, is significantly higher in the regions where BOLD variability increases from 33- to 40-wGA, namely the primary sensory (visual and auditory), sensorimotor and proto-DMN (precuneus and PCC). The expression of these genes remains significantly higher in these same regions during the late-fetal period, compared to regions without significant BOLD variability changes. Our findings support the idea that early transcriptional programs precede and lay the groundwork for later functional changes in the developing brain.

This prenatal regional upregulation of genes involved in neuronal ensheathment and gliogenesis coincides temporally and spatially with the functional and microstructural cortical maturation observed via MRI, characterized by longitudinal increases in BOLD variability, and reductions in cortical diffusivity and kurtosis.

A previous study in an independent dataset demonstrated a spatial alignment regarding cortical maturational differences between primary and higher-order cortical regions in term born infants, using MRI-based measures (T1/T2 ratio, cortical thickness, FA, MD, and NODDI intra-cellular volume fraction), reporting that the spatial patterning of cortical gene expression during late gestation was mirrored by a regional variation of cortical microstructure at term. More specifically, in comparison to higher-order areas, genes expressed in primary sensory areas at late gestation were found to belong to specific glial populations, including oligodendrocytes and microglia[46]. This aligns with our findings, denoting an increased expression of genes involved in neuronal ensheathment and gliogenesis in broadly the same regions, further elucidating mechanisms underlying the spatial

variation in brain cortical maturation captured by our functional and microstructural MRI data.

Despite the absence of histologic evidence for intracortical myelination in the human fetus and at the time of birth[87], it is possible that an increase in oligodendrocyte precursor cells (OPC) and early axonal cortical ensheathment by immature oligodendrocytes (O4+, O1+) may be at the origin of the observed decreases in cortical microstructural diffusivity. In fact, literature shows that OPC and immature oligodendrocytes produce mRNA levels of genes associated with myelination (such as MBP, PLP1, and MAG), as these cells prepare for differentiation, even before the myelin is detectable by immunostaining[88,89]. The decreases in kurtosis occurring during the same developmental period, especially in the radial direction, may reflect the retraction and disappearance of radial glia, an event that is known to coincide with the expansion of OPC during late gestation, marking a transition from neurogenesis to gliogenesis[90–92].

Furthermore, literature supports that the OPC and the early cortical axonal ensheathment by immature oligodendrocytes may both contribute to and be promoted by increased synaptogenesis[93–95]. This interplay could explain the concomitant increase observed in BOLD variability within the same cortical regions where the expression of these cells is increasing.

This analysis, therefore, provides important insights regarding the cellular processes underlying both the diffusion microstructural changes as well as functional changes observed with in vivo MRI data during this key period of brain development.

### Preterm birth impacts BOLD variability and cortical microstructure at TEA

Compared to FT newborns, very premature infants at TEA showed reduced BOLD-variability specifically in sensory RSNs (auditory and visual), the PCC, the thalamus, and the limbic, denoting the impact of preterm birth on cortical functional maturation. These results align with previous work, showing that BOLD variability, measured as ALFF (amplitude of low frequency fluctuations), is reduced in moderate-to-late preterm infants at TEA, compared to FT newborns, predominantly in the primary sensory and motor cortices, and in the posterior cingulate cortex and precuneus[32]. Our findings further complement recent evidence that the Hurst Exponential, which measures the temporal complexity of BOLD signals, is similarly reduced in somatosensory-motor, visual and auditory networks in preterm infants at TEA, compared to FT[55].

We also found that the cortical microstructure of VPT infants at TEA was characterized by overall increased microstructural diffusivity and kurtosis, across all RSNs cortical regions, further supporting that preterm birth affects not only functional, but also microstructural cortical RSNs maturation. Only a few studies have compared cortical microstructure between FT newborns and preterm infants at TEA using dMRI. Our findings align with the works of Ball et al., which demonstrated that, in comparison to FT, preterm infants at TEA present higher MD in frontal, parietal, occipital and temporal cortical regions[46,60]. We provide evidence of cortical kurtosis differences between these two groups at TEA, offering further insight into early microstructural development. VPT at TEA presented higher RK across all cortical RSNs regions, except for the thalamus, where there were no differences between groups. Interestingly, increases in MK were led by increases in RK, with no changes observed in AK between groups. Overall, our results suggest a delayed microstructural cortical maturation in VPT at TEA compared to FT.

As our study and others have shown, cortical diffusivity and kurtosis tend to decrease during brain development, until TEA[60,62]. As stated before, the longitudinal decrease in cortical diffusivity is thought to relate to an increase in dendritic arborization and density, as well as in gliogenesis, whereas the longitudinal decreases in kurtosis

may derive from the disruption of the radial glial scaffolding. These processes thus reflect ongoing cortical maturation, and appear to be disrupted by preterm birth, leading to increased cortical diffusivity and kurtosis at TEA in VPT compared to FT. In fact, cortical neurogenesis is mostly finished by mid-gestation, before 27-wGA, and it is followed by a shift to intense gliogenic events[81,91]. Preterm birth is known to expose the brain to noxious environmental exposures, conferring vulnerability during these important developmental processes. Prematurity may thus lead to a disruption of the ongoing gliogenic events, particularly oligodendrocyte maturation, which occurs during late gestation. In fact, hypoxic-ischemic events (a model of preterm injury) have been shown to damage precursors of oligodendrocytes[96,97]. In agreement with our findings and hypothesis, Ball et al. have shown that, compared to FT, preterm infants at TEA present, on average, lower cortical T1w/T2w contrast, and this contrast was found to correlate positively with the expression of genes associated with glial cells, including oligodendrocytes, during the second half of gestation[46].

Interestingly, the thalamus was the only region where no significant differences between groups were found for RK and MK. This may be due to the thalamus's distinct microstructural organization compared to the cortex, being more compact and exhibiting radial glia fibers mainly in the ventral nuclei only, which may reduce the sensitivity of kurtosis to detect group differences[98,99]. Nevertheless, group differences in diffusivity remained evident, with VPT at TEA presenting higher diffusivity in the thalamus than FT.

Despite the need of further research to better elucidate the mechanisms behind the increased cortical diffusivity and kurtosis observed in VPT at TEA in comparison to FT, our findings combining longitudinal dMRI data and genetic expression analysis during late gestation support that preterm birth may disrupt ongoing gliogenic and pre-myelination events, leading to an altered cortical microstructure by TEA.

This study also provides evidence of the effects of preterm birth on BOLD variability, which is decreased in VPT at TEA, namely in sensory RSNs (auditory and visual), the PCC, the thalamus, and the limbic network.

As stated, the rapid synaptogenesis during the late mid-fetal and early late fetal period is known to be heterochronous, following a spatiotemporal pattern, with earlier occurrence in primary motor and sensory areas, and latest in the prefrontal cortex[53]. It is possible that preterm birth, by disrupting the ongoing maturational processes, namely dendritic arborization and gliogenesis, will consequently impact synaptogenesis in the regions undergoing the most important maturational changes during late gestation, specifically primary sensory areas. Besides the visual and auditory, according to our findings, the PCC was also among the regions undergoing significant longitudinal increases in BOLD variability from 33 wGA to TEA, which may make it more vulnerable to preterm birth-related injury, in comparison to other cortical regions.

Previous studies evaluating the impact of preterm birth at TEA on functional connectivity have shown that VPT at TEA present a decreased RS-functional connectivity between the salience network and the auditory, visual, PCC and thalamic networks, compared to FT[43]. Other studies using fMRI and dMRI have also shown that the visual, PCC and thalamus are regions that are highly connected to each other and to others (rich-club nodes) and present decreased functional and structural connectivity in preterm infants at TEA compared to FT[41,100].

Decreased thalamo-cortical structural and functional connectivity has been consistently shown across studies in preterm infants at TEA compared to FT newborns[41,101,102]. In fact, thalamocortical connections are being established during the late second and third trimester of brain development, which is when preterm birth occurs[34,79]. By exposing the brain to noxious events, such as inflammation, hypoxia/ischemia and stress, preterm birth may affect the establishment of these connections. Disruption of thalamo-cortical connectivity may interfere with synaptogenesis in cortical primary sensory regions (such as the visual and auditory cortex), which rely heavily on the thalamic input for development, and are among the first to receive it. The PCC also receives thalamic input, although this occurs relatively later during gestation compared to primary sensory areas[103], which makes it a vulnerable target to the effects of preterm birth.

By linking our imaging findings, showing a disrupted structural and functional maturation in VPT at TEA, to the regional gene expression profiles from mid- to late-gestation, we provide deeper insights into the potential biological mechanisms underlying preterm birth injury. Specifically, gliogenesis and pre-myelination processes may be particularly vulnerable to disruption following preterm birth, contributing to the observed regional alterations in cortical microstructure and functional BOLD variability.

## Conclusion

Our findings support the biological significance of the brain's BOLD variability during early preterm infants' development, whose increase is accompanied by markers of regional cortical microstructural maturation and upregulation of genes mediating gliogenesis and premyelination events, specifically in primary sensory, sensorimotor and proto-DMN networks. Additionally, preterm birth appears to affect these maturational processes, being associated with reduced cortical functional and microstructural maturity at TEA, compared to FT birth.

By bridging genetic signatures with large-scale cortical organization, this work provides further insights regarding mechanisms underlying cortical maturation during late gestation and the putative mechanisms of preterm injury.

## Limitations

This study has limitations that should be considered. First, the sample size is modest. This limitation is largely related to the challenges associated with recruiting preterm infants and their families during a stressful time in their lives. The recruitment and execution of the project is further constrained by its longitudinal design, requiring two MRI scans, with an early first one by the 33rd wGA, followed by a second at TEA. This design necessitates the preterm infant to be clinically stable at 33 wGA, to undergo the MRI, which may not be always the case. Future studies with a larger, longitudinal, age-matched cohort will be important to strengthen the external validity of our findings.

Second, the spatial resolution limitations of both functional and diffusion imaging may affect the accuracy of the registration of the RS-fMRI ICA-based brain atlas—originally in the subjects' anatomical T2 space—to the functional or diffusion subjects' spaces. Such may introduce partial volume effects. However, since the pipeline is automated, any potential systematic errors would be consistent across subjects, allowing for valid longitudinal comparisons and between-groups analysis.

Third, the clinical significance behind brain BOLD variability and cortical microstructural alterations identified by DKI, DTI, and SMT metrics, following preterm birth, remains to be investigated. Evaluation of the long-term neurodevelopmental follow-up of this cohort during childhood is planned to investigate the relevance of these early imaging biomarkers.

Fourth, despite efforts to minimize motion during acquisition, neonatal MRI is inherently susceptible to motion artifacts, since no sedation was used; and despite application of motion correction analysis techniques, the quality of the data can still be affected by this limitation.

Fifth, the ICA-based brain atlas used in both our longitudinal and cross-sectional analyses was derived from a combination of VPT infants scanned at 3- wGA and at 40-wGA, as well as FT, representing our cohort of infants. Despite the variability of the brain and, possibly,

of the RSNs during this period, this approach was intentional, aiming to capture the variability of these RSNs during this period, which was considered essential for the longitudinal analysis. The same atlas was also applied to the cross-sectional analysis at term age, to ensure consistency in the regions evaluated, thereby facilitating direct comparisons with the longitudinal findings.

## Methods

### Subjects and study design

Research Ethics Committee approval was granted by the "Cantonal Research Ethics Committee of Geneva" and hosted at the University Hospitals of Geneva (HUG). Written parental consent was obtained prior to the infant's participation to the study.

Fifty-four VPT infants (<32 wGA at birth) and 24 FT infants were recruited at the neonatal and maternity units of the University Hospitals of Geneva (HUG), Switzerland, from 2017 to 2020.

Exclusion criteria for all babies included major brain lesions detected on the MRI, such as high-grade intraventricular hemorrhage, leukomalacia, as well as micro or macrocephaly or congenital syndromes.

For FT newborns, inclusion criteria included birth after 37 wGA, height, weight and head circumference above the 5th and below the 95th percentiles, APGAR score > 8 at 5 min and absence of resuscitation, infection or admission to the NICU (neonatal intensive care unit).

Six VPT infants were excluded from the study due to parental withdrawal or detection of a genetic anomaly, and two FT infants were excluded due to parental withdrawal.

Our study comprises both longitudinal and cross-sectional designs. The longitudinal design involves only our VPT infant cohort, who underwent MRI examinations at two time-points, firstly during the 33rd week of GA, and secondly at TEA. The cross-sectional design involves both the VPT at TEA and the FT newborns, who underwent a single MRI examination few days after being born (term age).

Infants whose MRI protocol acquisition was incomplete (not comprising a T2-weighted image, RS-fMRI sequence and multi-shell diffusion imaging (MSDI) sequence), without both longitudinal time-points (in preterm infants' case) or whose images presented excessive motion were excluded from the analysis. The final sample comprised 46 VPT infants and 21 FT newborns, of which 31 VPT and 19 FT newborns had data suitable for the RS-fMRI analysis, while 39 VPT infants and 18 FT newborns had data suitable for the diffusion microstructural analysis, with an overlap of 24 VPT infants and 16 FT newborns between the two analyses. The flow chart of the participant selection process is provided in Supplementary Fig. S5.

### MRI acquisition

All infants were scanned after receiving breast or formula feeding during natural sleep (no sedation used). At 33-wGA, preterm infants were scanned using an MR-compatible incubator (Lammers Medical Technology, Lübeck, Germany) and monitored using a Philips MR patient monitor Expression MR400 and Philips quadtrode MRI-compatible neonatal ECG electrodes. At TEA, infants were scanned using a vacuum mattress for immobilization. All infants, at both time points, were monitored for heart rate and oxygen saturation during the entire scanning time. MR-compatible headphones (MR confon, Magdeburg, Germany) were used to protect the infants from scanner noise.

MRI acquisition at both time-points was performed on a 3.0T Siemens Magnetom MR scanner (Siemens, Erlangen, Germany), using a 16-channel neonatal head coil. T2-weighted images were acquired using the following parameters: 113 coronal slices, TR = 4990 ms, TE = 160 ms, flip angle = 150°, matrix size = 256 × 164; voxel size = 0.8 × 0.8 × 1.2 mm³. RS-fMRI data acquisition was obtained by means of T2*-weighted gradient echo echo-planar imaging (EPI) sequence with the following parameters: 590 images, TR = 700 ms,

TE = 30 ms, 36 slices, voxel size = 2.5 × 2.5 × 2.5 mm³, flip angle = 60°, multiband factor = 4. MSDI was acquired with a single-shot spin echo echo-planar imaging (SE-EPI) Stejksal-Tanner sequence with the following parameters TE = 85 ms, TR = 3170 ms, voxel size 1.8 × 1.8 × 1.8 mm³, multi-band factor = 2, GRAPPA 2. Images were acquired in the axial plane, in anterior-posterior (AP) phase encoding (PE) direction, with 4 volumes without diffusion-weighting (b0); 10 noncollinear directions with $b$ = 200 s/mm²; 30 noncollinear directions with $b$ = 1000 s/mm²; 50 noncollinear directions with $b$ = 2000 s/mm². Additional b0 images were collected with reversed phase-encode blips, AP and posterior-anterior (PA), resulting in pairs of images with distortions going in opposite directions.

### RS-fMRI analysis pre-processing

The RS-fMRI data were preprocessed using SPM12 (Wellcome Department of Imaging Neuroscience, University College London, United Kingdom). The preprocessing steps included realignment to the mean functional volume, adjusting for motion, co-registration to the time-point specific structural image, alignment in Montreal Neurological Institute (MNI) space, and normalization, in which all the scans are warped to our cohort 40-wGA template. Slice-timing correction was omitted due to the fast TR of 700 ms used. Finally, the data were smoothed with a Gaussian filter of full width at half maximum (FWHM) of 5 mm. All volumes with a frame-wise displacement (Power, Mitra et al.[104]) greater than 0.5 mm or with a rate of BOLD signal changes across the entire brain (DVARS) greater than 3% were removed, along with the two previous and the two subsequent images. The remaining images were included for further analysis. A minimum of 50% of volumes remaining was set as sufficient criteria for inclusion.

### Resting-state atlas creation

RSNs were generated from our cohort of infants, comprising both VPT at 33- and 40-wGA, as well as the FT infants, and combined into a brain cortical parcellation atlas. To extract independent spatial networks, a group-level independent component analysis (group-ICA) was conducted; a z-score threshold of 2 was applied to these components. The results were acquired in a single group-level ICA using the GIFT toolbox in MATLAB (http://mialab.mrn.org/software/gift/index.html). Voxels in the cerebrospinal fluid, ventricles, eyes, and extracerebral areas (skull) were removed using a brain mask. The ICA was repeated 20 times using ICASSO for stability of the decomposition and determined 13 robust components when combining the 33- and 40-wGA scans, VPT and FT infants combined (Supplementary Fig. S6). Among the 13 components, 3 reflected areas related to motion and blood vessels, rather than resting-state activity. These were considered noise and excluded from further analyses. The 10 remaining components were combined into a brain atlas. This group-level ICA-based atlas was manually corrected to ensure anatomical precision. The cerebellum and the brainstem were removed from the final atlas, in order to keep only the cortical and subcortical structures for further analysis, which resulted in 9 RSNs: limbic (amygdala, insula, anterior cingulate cortex [ACC]), paralimbic belt (orbitofrontal cortex [OFC], temporal pole), thalamus, prefrontal cortex [PFC], visual, auditory, posterior cingulate cortex [PCC], sensorimotor, and precuneus (Fig. 5).

### Regional cortical BOLD variability estimation

To extract regional time series from the RS-fMRI data at each time-point, we used our ICA-based brain atlas consisting of 9 RSNs. First, the atlas was registered to each subject's native time-point specific space (structural T2-weighted) using the diffeomorphic symmetric image normalization algorithm with cross-correlation as similarity metric (SyN-CC), from the advanced normalization tools (ANTs)[105] toolbox.

Each subject's structural T2-weighted image was segmented using the tissue segmentation maps from the UNC neonatal AAL atlas[106],

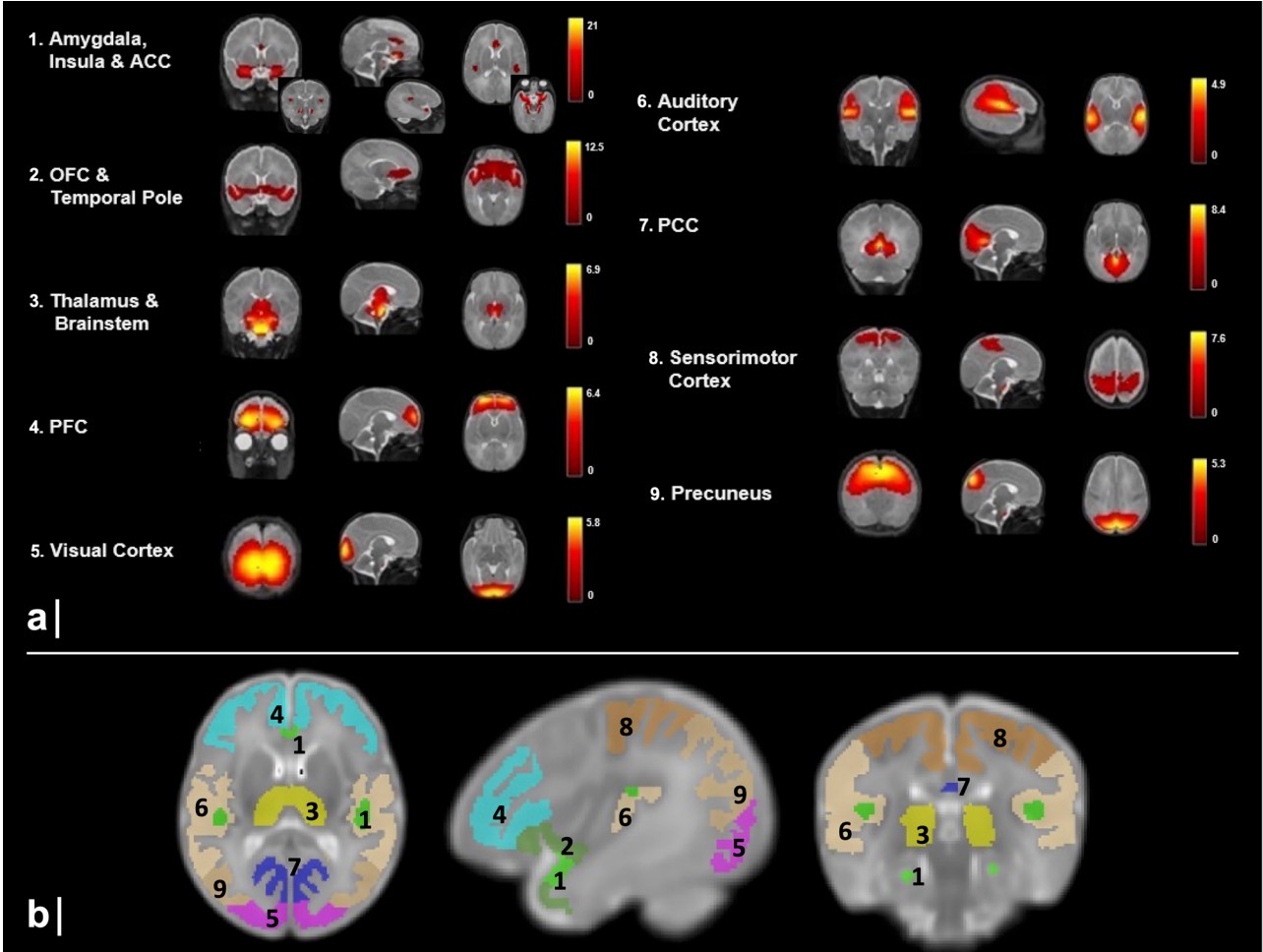

**Fig. 5 | Resting-state atlas. a** Main 9 components obtained from the data-driven ICA group analysis. Each row shows sagittal, coronal, and axial view of the components overlaid onto the 40 wGA template. A threshold at a z-score of 2 was applied. The colored bars show the corresponding z-score. **b** ICA-based brain atlas comprising 9 RSNs. RSNs are illustrated in the T2w 40 wGA template, after GM masking. 1. Limbic (amygdala, insula, anterior cingulate cortex), 2. paralimbic belt (orbitofrontal cortex, temporal pole), 3. thalamus, 4. prefrontal cortex, 5. visual, 6. auditory, 7. posterior cingulate cortex, 8. sensorimotor, 9. precuneus.

which include (GM) (cortical and deep GM), WM, and cerebral spinal fluid (CSF). These maps were first registered to the subject's native space using the same ANTS algorithm described above, to obtain the three tissue-probability maps (TPMs) in the native space. Next, using these TPMs, the subject's T2-weighted image was segmented using SPM12, so that subject-specific GM, WM, and CSF probability maps were obtained in the subject's T2 structural space.

To ensure that the BOLD signal was extracted only from GM voxels, the ICA-based brain atlas in the native space was GM-masked using the subject-specific TPMs obtained from the segmentation step. If the probability of one voxel being GM was less than the probability of this voxel being WM or CSF ($P(GM) < P(WM)$ or $P(GM) < P(CSF)$), then this voxel was not considered as GM and was excluded from the atlas.

The subject's GM-masked ICA-based atlas in the subject's structural space was then registered to the subjects' functional (BOLD) space, and the average regional BOLD time-courses were extracted, after performing voxel-wise nuisance regression (CSF, WM, motion signals), detrending and smoothing (Gaussian FWHM 5 mm), yielding a matrix with dimensions [#volumes, 9] for each subject and each RS-fMRI time-point (33 and 40 wGA).

The 9 regional time-courses were bandpass filtered ([0.01–0.1 Hz]) to discard noise components and non-resting state fMRI components. BOLD signal variability (BOLD SD) was estimated by computing the sample standard deviation of the time-course of each of the 9 RSNs. This step resulted in 9 regional BOLD variability values per subject and per time-point.

In addition, ALFF (amplitude of low-frequency fluctuations) was computed by transforming each regional time series into the frequency domain using the Fast Fourier Transform to obtain the power spectrum[107–109]. The square root of the power spectrum within the 0.01–0.08 Hz (low) frequency band was averaged to obtain regional ALFF values. Regional fALFF (fractional ALFF) was then calculated as the ratio of the low-frequency (0.01–0.08 Hz) ALFF to the total ALFF across all frequencies of the power spectrum. These complementary measures were used to confirm that the results of BOLD SD analyses of spontaneous BOLD fluctuations were consistent across other different metrics estimating BOLD variability.

We selected BOLD variability (BOLD SD) as the primary measure because it captures moment-to-moment fluctuations across the full frequency spectrum, providing a broad and sensitive index of spontaneous neural dynamics. In contrast, ALFF and fALFF are restricted to a predefined low-frequency band and are more dependent on pre-processing choices, whereas BOLD variability provides a more general, assumption-free characterization of spontaneous activity, which is better suited to capturing physiological and microstructural changes

during late gestation. ALFF and fALFF results are presented in the Supplementary Material.

## Multi-shell diffusion imaging pre-processing

MSDI data were preprocessed using MRtrix3 (version 3.0rc3, https://www.mrtrix.org) (Tournier et al.[110]) for denoising, bias field corrections and intensity normalization. The FSL diffusion toolbox (v5.0.11, https://fsl.fmrib.ox.ac.uk/fsl/fslwiki/) (Behrens et al.[111], Smith et al.[112]) was used for the preprocessing of MSDI data. FSL's TOPUP (Andersson et al.[113], Smith et al.[112]) was used to estimate the off-resonance field, which was then used as input for FSL's EDDY function optimized for neonatal diffusion data, correcting for distortions induced by susceptibility and eddy currents, as well as by motion-induced signal dropout and intra-volume subject movement (Andersson et al.[114], Andersson et al.[115], Bastiani et al.[116]). Data were visually inspected to assure the quality of motion artifacts correction.

## Diffusion cortical microstructural maps

After pre-processing, MSDI data were analysed using three different methodological approaches, each applied to the same set of $b$-values ($b = 200$, $b = 1000$, $b = 2000$ s/mm$^2$).

To examine cortical microscopic features unconfounded by the directional structure, we use the SMT[30], which focus on the direction-averaged diffusion weighted signal, independent of the fiber orientation distribution. MSDI was analysed using the multi-compartment microscopic diffusion model from the publicly available SMT code (https://github.com/ekaden/smt). SMT metrics corresponding to intra-neurite volume fraction (intra), intrinsic diffusivity (Diff), extra-neurite mean diffusivity (extraMD) and extra-neurite transverse diffusivity (extraTrans) were calculated.

Alongside, we calculated DTI metrics (FA; MD), measuring Gaussian diffusion properties of the water in the tissue, as well as DKI metrics (mean/radial/axial kurtosis, MK/RK/AK), evaluating non-Gaussian diffusion and allowing the characterization of tissue microstructural heterogeneity[27,117]. For both DTI and DKI metrics, we used the code publicly available from the DIPY library (https://docs.dipy.org/stable/interfaces/reconstruction_flow.html), applying the DKI model to the full MSDI data.

The GM-masked ICA-based brain atlas in the subject's structural space was registered to the subjects' dMRI space using ANTs[105], and averaged diffusion metrics were extracted per cortical RSN and per subject, at each time-point.

## Gene expression analysis

Regional patterns of gene expression in the fetal cortex across gestation were obtained from the BrainSpan developmental transcriptomic dataset (http://development.psychencode.org/), comprising mRNA-seq collected from $n = 41$ specimens aged between 8 pcw and 40 postnatal years. For our study, we have included data sampled from 18 post-mortem prenatal brain specimens, aged 8–37 pcw ($n = 165$ regional tissue samples, mean [SD] age = 17.74 [7.54] pcw, 53% male, mean [SD] postmortem interval = 6.52 [11.03] h, mean [SD] RNA integrity number [RIN] = 9.34 [0.66]).

Tissue had been collected after obtaining parental or next of kin consent and with approval by the institutional review boards at Yale University School of Medicine, the National Institutes of Health, and at each institution from which tissue specimens were obtained. For each brain, regional dissection including 11 neocortical regions was performed (dorsolateral PFC [DLPFC], ventrolateral PFC [VLPFC], OFC, medial frontal cortex [MFC], primary motor/sensory/auditory/visual cortex [M1, S1, A1C, and V1], inferior parietal cortex [IPC], superior temporal cortex and inferior temporal cortex). Tissue processing and detailed anatomical boundaries for each cortical region at each stage of development are provided elsewhere[86,118]. Regional tissue samples were subject to mRNA-seq using an Illumina Genome Analyzer IIx

(Illumina, San Diego, California, United States of America) and mRNA-seq data processed using RSEQtools (version 0.5)[119]. Gene expression was measured as RPKM. Conditional quantile normalization was performed to remove guanine-cytosine (GC) content bias, and ComBat was used to remove technical variance due to processing site (Yale or University of Southern California)[118,120,121].

The prenatal gene expression data were initially filtered to only include protein-coding genes (NCBI GRCh38.p12, $n = 18,524$ out of a possible 20,720). In order to restrict our analysis to focus on genes expressed in the developing cortex, this list was further filtered to only contain genes expressed by cells in the fetal cortex based on the composite list of prenatal cell markers from 5 independent single-cell RNA studies of the developing fetal cortex[46]. This resulted in expression data from a final set of 5287 genes.

For our study, gestational periods were grouped into early-fetal (T0, 8–13 pcw), mid-fetal (T1, 16–22 pcw) and late-fetal (T2, 35–37 pcw). The neocortical regions with available genetic data were matched to our RSNs as follows: MFC = limbic (1), OFC = paralimbic belt (2), DLPFC + VLPFC = PFC (4), V1 = visual (5), A1C = auditory (6), M1 + S1 = sensorimotor (8), IPC = precuneus (9). Next, these regions were grouped according to the presence (group 1) or absence (group 2) of significant BOLD variability changes from 33- to 40-wGA (Fig. 3a, b).

## Neonatal and demographic data statistical analysis

To test for differences in demographic and perinatal data between VPT and FT groups, categorical variables (sex, intrauterine growth restriction, sepsis, and intraventricular hemorrhage grade I) were analysed using chi-squared test, whereas continuous variables were compared using independent samples t-test, with group as an independent variable and the following dependent variables: GA at birth, GA at MRI at TEA, birth weight, birth height, birth head circumference, APGAR score at 1 and 5 min after birth and parental socio-economic status score (SES)[44].

## MRI data statistical analysis

Longitudinal (from 33- to 40-wGA) and cross-sectional differences (at TEA, between VPT at 40-wGA and FT newborns) in both cortical BOLD variability and microstructural diffusivities were assessed using two-sided paired-samples t-tests and independent samples t-tests, respectively, with FDR correction for multiple comparisons. Prior to statistical testing, outliers were identified for each metric, by region and timepoint, based on 1.5 × interquartile range (IQR). Values falling below Q1 − 1.5 × IQR or above Q3 + 1.5 × IQR were excluded from subsequent analyses. All analyses and visualizations were performed using RStudio (R version 4.4.1).

For visualization of inter-individual variability using boxplots, all subject-level data were normalized by the maximum absolute change across all participants and regions for each metric, thereby constraining values to the range [−1, 1] while preserving the directionality of effects. These globally normalized values were used to generate boxplots, enabling comparison of relative regional patterns across participants and across metrics, independent of absolute effect magnitude. Additional boxplots displaying the raw, non-normalized data are provided in Supplementary Material (Fig. S2).

For bar-plot visualizations and the clustering analysis, for each metric, mean regional changes were first computed for each cortical resting-state network region and then normalized per metric by dividing by the largest absolute mean regional change, preserving the directionality of effects. This scaling yielded a group-level region × metric matrix representing the relative spatial distribution of longitudinal changes across cortical resting-state network regions.

For Lasso-based regression modeling, the same normalized group-level region × metric matrix was used (mean regional changes were first computed for each cortical resting-state network region and then normalized per metric by dividing by the largest absolute mean

regional change), but expressed as absolute values, in order to focus on the magnitude of change, irrespective of the direction. This approach prevents potential cancellation of positive and negative effects across regions and ensures that predictors with strong effects in opposite directions are appropriately retained.

Microstructural diffusivities were clustered using the R-package and function ConsensusClusterPlus (k-means base algorithm and Euclidean distance)[122], to identify stable clusters of regional diffusivities. The optimal number of clusters was determined using the elbow method, based on the within-cluster sum of squares. This approach enabled grouping cortical regions based on the similarity of their longitudinal microstructural maturation, and to compare these microstructural clusters with those derived from BOLD variability longitudinal changes.

Lasso-based sparse multivariate regression models were used to identify the most relevant microstructural predictors of BOLD variability longitudinal changes, between 33- and 40-wGA. Analyses were performed using the glmnet package in R with leave-one-region-out cross-validation. The analysis was conducted on the group average 9 region × 10 metric matrix ($n = 9$ regions), where BOLD variability change served as the dependent variable and nine microstructural metrics (extraMD, extrTrans, diff, intra, MD, FA, MK, AK, and RK) served as predictors. All metrics were standardized prior to modeling, as previously described. Model selection was performed using leave-one-region-out cross-validated mean squared error, and for each penalty value, RMSE and $R^2_{loo cv}$ were calculated. Two penalty choices were evaluated: the more conservative penalty corresponding to one standard error above the minimum ($\lambda_1 se$) and the less penalized minimum-error solution ($\lambda\_min$). In addition to a pure LASSO model ($\alpha = 1$), a LASSO elastic net model ($\alpha = 0.5$) was examined to allow for partial retention of correlated predictors. Predictors with non-zero coefficients at the selected penalty values were considered relevant.

### Gene expression data analysis

A linear mixed effects regression model (LMM) was used to identify significant changes in gene expression as a function of regional group (1 and 2) and time, with fixed effects of sex and RIN (RNA integrity number), including specimen ID as a random effect. We identified genes with a significant group *time interaction and calculated estimated marginal means, followed by pairwise comparisons to assess differences in gene expression over time and between groups. Multiple comparisons were adjusted using FDR correction. Modeling was performed in RStudio (R version 4.4.1) using the "lmer()" function from the lme4 package. Gene function was classified using gene set enrichment analysis using WebGestalt[45]. Enrichment ratios were calculated as the proportion of cell class-specific genes in our gene list of interest compared to the proportion in the full background set. The background gene set was defined as the full list of protein-coding genes included in the analysis ($n = 5287$ expressed by cells in the fetal cortex). We corrected for multiple comparisons across cell classes using FDR. The genes with a significant group*time interaction, found to have a significant enrichment ratio linked to neocortical organization, were combined for expression analysis across time points (T0, T1, and T2) and between groups (1 and 2) using two-way ANOVA followed by post-hoc comparisons using Tukey's test (HSD).

### Reporting summary

Further information on research design is available in the Nature Portfolio Reporting Summary linked to this article.

## Data availability

All neuroimaging data were acquired in the context of a research project approved by the ethical committee in 2016. The raw data are protected and are not available due to data privacy laws. Developmental RNA-seq data used in this study were downloaded from: http://development.psychencode.org/. Source data are provided with this paper.

## Code availability

Software and code used in this study for MRI analysis are publicly available as part of FSL v5.0.10 (https://fsl.fmrib.ox.ac.uk/fsl/fslwiki/), MRtrix3 (Tournier et al.[110]), SMT and DIPY software packages. dMRI data were pre-processed using the EDDY command adapted for neonatal motion, from the neonatal dMRI automated pipeline from the developing Human Connectome Project (dHCP, http://www.developingconnectome.org), and can be found at: https://git.fmrib.ox.ac.uk/matteob/dHCP_neo_dMRI_pipeline_release (Bastiani et al.[116]). Supporting code for this manuscript, used to generate the results and figures is available on Zenodo (https://doi.org/10.5281/zenodo.18875986).

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

## Acknowledgements

The authors thank all clinical staff, namely in the neonatology and in the unit of development of the HUG Children's Hospital, all parents and newborns participating in the project, the Pediatrics Clinic Research Platform and the Center for Biomedical Imaging (CIBM) of the University Hospitals of Geneva, for all their valuable help and support. This study was supported by grants from the Swiss National Science Foundation (no. 32473B_135817/1 and no. 324730–163084, J.S.A., A.B., and L.L.), the Prim'enfance Foundation (P.S.H., L.L.), the foundation ART-THERAPIE (P.S.H., clinical staff), the Swiss Government Excellence Scholarship (no. 2017.0450/OP, J.S.A.), the Swiss Academy of Medical Sciences (YTCR 49/19, J.S.A.), the "Fondation pour la recherche en périnatalité" (P.S.H., clinical staff), and an NHMRC Investigator Grant (APP1194497, G.B.).

## Author contributions

J.S.A.: data collection and investigation, conceptualization and design of methods and analysis, formal analysis, writing-original draft. A.B.: formal analysis, design of methods and analysis; writing-review and editing. S.L.: conceptualization and design of methods and analysis, writing-review and editing. E.F.: conceptualization of methods, writing-review and editing. A.V.: data analysis, writing-review and editing. L.L.: data analysis, writing-review and editing. S.C.: resources, investigation. F.L.: resources, investigation. D.V.V.: conceptualization and design of methods and analysis; writing-review and editing. G.B.: conceptualization and design of methods and analysis, supervision, resources, writing-review and editing. P.S.H.: conceptualization, supervision, resources, funding acquisition, writing-review and editing.

## Competing interests

The authors declare no competing interests.

## Additional information

**Supplementary information** The online version contains Supplementary material available at https://doi.org/10.1038/s41467-026-71415-x.

