## [Transparent Peer Review File · Nature Communications]

Regional BOLD variability reflects microstructural maturation and neuronal ensheathment in the preterm infant cortex

Corresponding Author: Dr Joana Sa de Almeida

Version 0:

Reviewer comments:

Reviewer #1

(Remarks to the Author)

The non-invasive imaging readout - BOLD signal variability - indirectly reflects spontaneous neuronal activity and varies across cortical regions, cognitive tasks, and age. This manuscript aims to investigate how BOLD variability evolves in very preterm (VPT) infants, how it correlates to cortical microstructure and gene expression, and how it differs from full-term (FT) newborns at term-equivalent age (TEA).

To achieve this, the authors collected resting-state fMRI and multi-shell diffusion imaging for VPT infants (at 33 and 40 weeks GA) and FT infants (40 weeks GA). They first compared BOLD variability longitudinal changes in VPT infants between 33 and 40 weeks GA. Primary sensory, sensorimotor and pDMN regions (cluster 1) had increased BOLD variability across 33 to 40 weeks. The authors then correlated longitudinal changes in BOLD variability with cortical microstructural changes as measured by diffusion imaging metrics. In regions where BOLD variability increased with GA, diffusivity decreased – revealing a relationship between cortical microstructure and BOLD variability. Next, the authors associated longitudinal microstructural and BOLD variability changes with gene expression changes using the brainspan developmental transcriptome. This analysis revealed that cluster 1 regions had increased gene expression profiles associated with neuronal ensheathment and gliogenesis. Lastly, they compared BOLD variability and cortical microstructure at term equivalent age between VPT and FT infants revealing regions-specific decreases in BOLD variability and global increases in diffusivities in VPT infants at TEA.

Overall this study uses state-of-the-art methodologies to uncover the biological significance of BOLD variability changes in the early development of VPT infants. This is an important topic as the field would benefit from a better understanding of underlying mechanisms of preterm birth-induced brain injuries. However, the study has several issues that should be investigated and resolved.

1) The applicability of fetal gene expression patterns associated with cluster 1 regions to the longitudinal changes in BOLD variability is difficult to interpret. Surely, there is an increase in gene expression associated with gliogenesis and myelination from T1 to T2, however, this is a much broader developmental timeframe (16-22GA to 35-37GA) than is assessed with imaging (33GA to 40GA).

2) Gene expression was analyzed using a filtered list of protein-coding genes that were identified in single-cell datasets of the fetal cortex. Single-cell transcriptomic techniques often have low sensitivity and do not fully capture RNAs expressed at low levels. The filtering performed here greatly limits the potential to identify novel biological processes that may be underlying physiological changes identified here.

3) The novelty and overall significance of these findings are difficult to assess without contextual discussion of two recent studies examining resting-state fMRI signals in preterm infants (Drayne et al., 2024 & Mella et al., 2024).

4) TEA and FT imaging data is taken from infants that have been exposed to postnatal environments for ~7 weeks vs. a few days, respectively. The analysis of BOLD variability and microstructural differences between these groups do not consider the role of exogenous factors that may influence their interpretation of the data.

Reviewer #2

(Remarks to the Author)

This manuscript aims to characterize changes in BOLD variability and their structural and biological underpinnings in very preterm (VPT) infants, with comparisons to full-term (FT) infants. The authors used fMRI to assess regional BOLD variability, diffusion MRI to evaluate cortical microstructure across longitudinal preterm data (33w GA to TEA), and independent gene expression analyses to explore biological correlates of BOLD variability. A LASSO model was applied to investigate the relationship between microstructure and functional variability. It is very interesting to study the difference between VPT and FT from the perspective of BOLD signal variability and its relationship with cortical microstructure and gene expression. Addressing the interplay between BOLD variability from resting-state fMRI, microstructure measures from diffusion MRI, and gene expression is a major strength of this work. However, there are notable concerns about the fundamental mechanisms of BOLD variability and rigor of methodology, which limit the impact and robustness of the current work.

Major Comments

1. The fundamental mechanisms underlying BOLD variability are not clear. Thus, it does not appear robust to establish a study centered on a measure with its mechanism not well understood. The limited citations provided to explain BOLD variability are constrained to few groups, raising the question if BOLD variability is broadly reproducible or widely applicable.
2. Methodology needs to be further improved to be more rigorous. For example, the description of the LASSO model is relatively vague. It is not clear how the regression was performed, how significance was determined, or what the model performance was. In addition to listing the selected three measures, the manuscript should report the actual sample size, model fit statistics, and p-values for all predictors. As another example, the analysis of cortical gray matter microstructure using diffusion MRI should consider mitigating partial volume effects.
3. The relatively small sample size, combined with the lack of validation on publicly available datasets, raises concerns about the generalizability of the findings. The results would be more convincing with rigorous validation. Although the authors note the difficulty of acquiring longitudinal preterm participants, the Developing Human Connectome Project (dHCP) contains a large cohort of both preterm and term infants, including longitudinal scans. Leveraging this public dataset would greatly strengthen the conclusions.
4. While the authors chose 9 microstructure measures from diffusion MRI, BOLD variability was used as the only measure derived from resting state fMRI. Other relatively more widely adopted measures from resting-state fMRI such as amplitude of low frequency fluctuations (ALFF) or regional homogeneity (ReHo) should be included to test the findings or the authors should justify exclusion of these other more widely used measures from resting state fMRI. Lacking these measures from resting state fMRI weakens the functional analysis and limits the ability to fully address how microstructural and biological development support functional maturation.

Minor Comments

1. While the Introduction provides useful background on diffusion models, it does not sufficiently justify why DTI measures (e.g., FA, MD) were applied to study cortical microstructure, given their traditional use in white matter.
2. Since the authors used three different diffusion models, it would be beneficial to specify the b-values used in each diffusion model in the Methods.
3. The definition of "clusters" is not clear. In BOLD variability, "clusters" appear to denote groups of regions showing significant increases between two timepoints. But in diffusion MRI, k-means clustering (k=3) was used without justification of why three clusters were chosen. To avoid confusion, the authors could consider using "groups" for the BOLD variability groups of regions and provide validation (e.g., elbow method, silhouette score) for the k-means choice of clusters.
4. The authors might want to be cautious about using "trajectory", as only two timepoints were included in the manuscript.
5. There is a typo in the last sentence of the Introduction: "weather" should be corrected to "whether."

Reviewer #3

(Remarks to the Author)

Thank you for asking me to review this interesting paper examining neonatal brain development.

The work combines MRI data with open access transcriptomic data to produce interesting and plausible biologic insights. The combination of BOLD as a measure of cortical function with transcription data is as far as I know completely novel and of great interest. The combination of MRI with transcription data is an emerging and promising area of science and Ball is the world leader in developing this approach in the developing brain. The paper leverages longitudinal imaging data encoding the 'experiment of nature' that is preterm delivery. Premature birth is a wide-ranging and robust disrupter of brain development, which has been shown (by these authors as well as many others) to affect almost every aspect of brain development and so provides a useful 'positive control' reality check on these results.

The imaging data appropriately collected with reasonably up to date sequences. The cohort is small, and indeed quite a lot smaller than most substantial studies currently being published. The data were collected by the researchers, and they probably trust their own data quality, relying on the longitudinal data to add value to the small cohort. Nevertheless, in the same way that they used open access transcriptomic data, they could have replicated their results using the open access imaging data now freely available, which are an order of magnitude larger than the current dataset, and which some of this group have used in the past.

The image analysis is reasonable. It appears that there was no motion correction applied to the scan data, but a reasonable

cut-off for exclusion of scans was used to remove corrupted datasets. It is a bit controversial whether this scrubbing of individual data (which disrupts the time course) affects fMRI interpretation, but my view is that it is unlikely to invalidate measures of BOLD variability. Variability is a nice and under-explored measure of cortical function, the results are interesting and as far as I know novel, and the functional locations revealed are interesting. Spherical means is a good method for analysing diffusion data and the results are consistent with previous studies using different approaches.

The imaging results are plausible and interesting, with the increases in visual and sensory-motor cortex in late gestation as expected given known developmental pathways, and it is also interesting that they did not find BOLD variability increases in pre-frontal cortex or limbic- it is plausible to interpret this as the frontal cortex not having started its rapid development while the limbic system may have completed its comparable phase. However it would have been interesting to see more data on temporal cortex which is rapidly developing at this time.

The transcriptomic data and analysis are high quality and appropriate. It is helpful to have the three time points in the data even if there are no imaging data to compare to the first period. The gene set analysis is uncontroversial and Lasso is a reasonable approach to this underdetermined dataset. As with the imaging data the results are plausible and it is reassuring that neuronal ensheathment is most significant relation, which gives confidence in other pathways detected. It is a shame that the analysis does not make use of the remarkable microbrain histologic parcellation that one of the authors has recently published, but relies on small patch location for the transcriptomic analysis, but that does not invalidate the results. The gene sets exposed in the analysis are interesting and add to existing knowledge.

Version 1:

Reviewer comments:

Reviewer #1

(Remarks to the Author)

I thank the authors for carefully addressing my comments. I believe the updated version of this manuscript is much improved and will be of great interest to the field.

Reviewer #2

(Remarks to the Author)

It is well appreciated that the authors have done a great job in revising this manuscript. The authors have addressed previous comments by incorporating additional and relevant references on BOLD variability, conducting further analyses using other BOLD signal measures, providing more detailed methodological descriptions, and discussing the limitations related to the relatively limited dataset. The manuscript has been improved significantly after revision. Just a couple of minor comments remain as follows.

1. It would be helpful to clarify in the Methods section whether all features included in the LASSO regression were standardized prior to model fitting. Given the different scales of the diffusion-derived measures, explicit reporting of normalization is important. In addition, several of the microstructural metrics are likely highly correlated; therefore, the selection of MD as a predictor should be interpreted with caution. The authors may consider toning down language suggesting MD as the "best" predictor and instead emphasize that LASSO identifies representative features from correlated sets.

2. The Discussion cites prior studies applying NODDI to cortical gray matter. Since NODDI was established for white matter modeling and does not account for somas, making its application to cortical gray matter problematic. Previous studies tailored for cortical gray matter microstructure (e.g. PMID: 32289460; PMID: 41486548) have repeatedly found that NODDI might not be used appropriately for quantifying cortical microstructure. Considering these specific works, the authors should briefly acknowledge the methodological limitations and interpret NODDI-related findings with caution.

Reviewer #3

(Remarks to the Author)

The authors have addressed my questions.

REVIEWER COMMENTS

We thank sincerely the reviewers and the editor for their constructive comments and suggestions that have greatly improved our manuscript. Detailed responses are provided below in blue, and additions to the manuscript are highlighted in yellow.

Reviewer #1:

Overall, this study uses state-of-the-art methodologies to uncover the biological significance of BOLD variability changes in the early development of VPT infants. This is an important topic as the field would benefit from a better understanding of underlying mechanisms of preterm birth-induced brain injuries. However, the study has several issues that should be investigated and resolved.

- 1. The applicability of fetal gene expression patterns associated with cluster 1 regions to the longitudinal changes in BOLD variability is difficult to interpret. Surely, there is an increase in gene expression associated with gliogenesis and myelination from T1 to T2, however, this is a much broader developmental timeframe (16-22GA to 35-37GA) than is assessed with imaging (33GA to 40GA).**

We agree with the reviewer that the developmental time frame is not exactly the same between the fetal gene expression and our neuroimaging data. The available developmental transcriptomic dataset (<http://development.psychencode.org/>) is limited by the inherent difficulty in obtaining well-characterized human fetal brain samples across the full gestational period. It provides data sampled from 18 post-mortem prenatal brain specimens aged 8-37 pcw, which we have clustered as proposed in (Kang et al., 2011), according to developmental windows. Since only two specimens fall within the specific developmental window in which our cohort was scanned (Figure R1A), excluding samples from the broader prenatal period would prevent us from modelling the temporal gene-expression trends occurring during our observational window. We would also like to clarify that, as we expect gene expression changes to precede structural and functional changes, the available timeframes align well with the aims of our study, allowing us to model change in gene expression that occur both before and during the observed regional differences in functional and structural maturation in the cortex.

Our results are consistent with the findings from (Kang et al., 2011), illustrating the well-described temporal sequence in cortical gene expression, with cell proliferation genes peaking first, followed by expression of immature neuron markers (e.g., DCX, marker of neurogenesis and neuronal migration), and subsequently gliogenesis and myelination gene programs (Figure R1C, (Kang et al., 2011)). The expression of genes involved in cortical myelination increases well before myelin becomes detectable on histology, which is typically after birth (Figure R1B, (Miller et al., 2012)). In line with Kang et al., we find a marked increase in myelination-related gene expression beginning from the mid-fetal period (100 days post-conception \approx 16wGA) and continuing until term age. We show that from 33 to 40 wGA, while myelination-related gene expression is rising, regions exhibiting the largest increases in BOLD variability in our dataset

correspond to those with a significantly higher expression in those genes. Plus, the expression of these genes remains significantly higher in these same regions during the late-fetal period, compared to regions without significant BOLD variability changes. These findings support the idea that early transcriptional programs lay the groundwork for later functional changes.

We have added this rationale into the Discussion, section “Genes mediating gliogenesis and neuronal ensheathment increase more in RSNs where BOLD variability increased”, as follows:

Between 16-22 pcw and 35-37 pcw, corresponding to the mid- to late-fetal period, analysis of ex-vivo fetal cortical gene expression patterns revealed a significant upregulation of genes associated with neuronal ensheathment and gliogenesis. **These findings are consistent with previous literature showing a marked increase in myelination-related gene expression starting from the mid-fetal period, continuing until birth, and into post-natal period (Kang et al., 2011).**

This increase in genes associated with neuronal ensheathment and gliogenesis, from mid- to late-fetal period, is significantly higher in the regions where BOLD variability increases from 33- to 40-wGA, namely the primary sensory (visual and auditory), sensorimotor and proto-DMN (precuneus and PCC). The expression of these genes remains significantly higher in these same regions during the late-fetal period, compared to regions without significant BOLD variability changes. **Our findings support the idea that early transcriptional programs precede and lay the groundwork for later functional changes in the developing brain.**

Figure R1. A. Bar plot showing the number of observations (specimen ID) per gestational age (in weeks), obtained from the BrainSpan developmental transcriptomic dataset (<http://development.psychencode.org/>). B. Adapted from (Miller et al., 2012), bar graph depicts mean percent of maximum mature adult myelinated fiber length density (MFLD) across development in humans, across cortical regions. C(d). Adapted from (Kang et al., 2011), sets of genes (expressed as percentage of maximum) plotted against age to represent general trends and regional differences in several neurodevelopmental processes in NCX (neocortex), HIP (hippocampus) and CBC (cerebellum).

- 2. Gene expression was analyzed using a filtered list of protein-coding genes that were identified in single-cell datasets of the fetal cortex. Single-cell transcriptomic techniques often have low sensitivity and do not fully capture RNAs expressed at low levels. The filtering performed here greatly limits the potential to identify novel biological processes that may be underlying physiological changes identified here.**

We thank the reviewer for their comment. We agree that single-cell transcriptomic datasets may have reduced sensitivity for low-abundance transcripts. We emphasize that BrainSpan (<http://development.psychencode.org/>) which provides bulk mRNA-seq data, was used as the primary dataset, and that the single-cell datasets were used solely to define a high-confidence list of genes expressed in the developing cortex, not as the source of the expression profiles analyzed in our study.

The BrainSpan bulk mRNA-seq dataset was generated using standardized protocols with stringent quality-control criteria, aiming to minimize false positives and capture robust region-level transcriptional patterns. Our filtering strategy follows standard practice for restricting bulk transcriptomic analyses to genes with validated prenatal cortical expression, using high-confidence markers derived from multiple independent single-cell datasets, to ensure that our analysis targeted biologically meaningful cortical signals. This consensus reduces noise from transcripts not expressed in the fetal cortex, increases specificity for robust developmental processes, and minimizes false positives introduced by bulk-level technical variation. Our aim was to characterize developmental patterns in genes with validated relevance to fetal cortical biology, rather than to conduct an exhaustive exploratory screen for novel low-abundance transcripts.

The developmental trajectories identified with this biologically grounded gene set are consistent with previous literature (Miller et al., 2012) and, importantly, align with the functional and structural neuroimaging changes observed during this period, indicating that the major biological processes are robustly captured despite this conservative filtering.

- 3. The novelty and overall significance of these findings are difficult to assess without contextual discussion of two recent studies examining resting-state fMRI signals in preterm infants (Drayne et al., 2024 & Mella et al., 2024).**

We thank the reviewer for highlighting these interesting recent studies. Indeed, Drayne et al. (2024) and Mella et al. (2024) examine resting-state fMRI signals in preterm infants using the Hurst Exponent (HE) to characterize the temporal complexity of BOLD signals. While both HE and BOLD variability (BOLD SD) capture aspects of neural signal dynamics, the two metrics reflect distinct properties. BOLD variability (BOLD SD) specifically quantifies the moment-to-moment fluctuations in signal amplitude, providing a measure of signal variability sensitive to region-specific functional activity. In contrast, HE captures temporal

correlations over longer time scales, aiming to characterize the temporal complexity of BOLD signals, indexing the persistence or “memory” (how past fluctuations of the signal predict future fluctuations) of the signal, rather than its instantaneous amplitude variability.

As suggested, we have added to our discussion the results from Drayne et al. (2024) and Mella et al. (2024) studies, adding these independent but related perspectives on functional maturation in preterm infants.

Discussion, Section “Cortical microstructural maturation underlies RSNs BOLD variability increases during preterm infants’ brain development”:

Our findings, revealing significant longitudinal increases in BOLD variability specifically in primary motor, sensory, and posterior brain regions from 33wGA to TEA, with no changes in prefrontal regions, agree with the spatiotemporal pattern of synaptogenesis and cortical activity. These results, quantifying fluctuations in BOLD signal amplitude, complement previous literature findings showing that the temporal complexity of BOLD signals, evaluated by means of the Hurst Exponent (HE), also increases overall in very preterm infants’ gray-matter from birth until TEA (Drayne et al., 2024; Mella et al., 2024). Similar to our findings, the greatest increases in HE were also found in motor and visual networks, compared to frontal networks, suggesting that primary sensorimotor areas mature from “more random” to “more ordered” states faster than the frontal ones (Drayne et al., 2024). BOLD variability provides thus a complementary perspective, suggesting that primary sensorimotor regions also present increasingly variable functional activity, highlighting their accelerated functional maturation.

Discussion, Section “Preterm birth impacts BOLD variability and cortical microstructure at TEA”:

Compared to FT newborns, very premature infants at TEA showed reduced BOLD-variability specifically in sensory RSNs (auditory and visual), PCC, thalamus and limbic, denoting the impact of preterm birth on cortical functional maturation. Our findings further complement recent evidence that the Hurst Exponent, which measures the temporal complexity of BOLD signals, is similarly reduced in somatosensory-motor, visual and auditory networks in preterm infants at TEA, compared to FT (Mella et al., 2024).

- 4. TEA and FT imaging data is taken from infants that have been exposed to postnatal environments for ~7 weeks vs. a few days, respectively. The analysis of BOLD variability and microstructural differences between these groups do not consider the role of exogenous factors that may influence their interpretation of the data.**

We thank the reviewer for raising this point. The primary aim of our analysis was to characterize whether VPT infants at TEA (and thus exposed to a postnatal environment early) differ from FT infants in cortical BOLD variability and microstructure. Because our goal was to characterize group-level differences rather than infer causal relationships, we applied inferential statistics without covariates. Importantly, early postnatal environmental exposure is an intrinsic

feature of preterm birth. Adjusting this analysis for additional specific clinical exposures of the VPT group might inadvertently remove biologically meaningful variance that is not necessarily confounding to the contrast of interest.

We would also like to highlight that our VPT cohort was relatively homogeneous: included only infants born before 32 weeks of gestation who were clinically stable and free of major complications. Notably, there were no statistically significant differences between the VPT and FT groups in demographic and clinical variables such as sex, socioeconomic status (SES), intrauterine growth restriction (IUGR), bronchopulmonary dysplasia (BPD), or intraventricular hemorrhage (IVH). The GA at the MRI scan was also similar (Table 1). This strengthens the interpretability of the observed group differences.

While the range of potential exogenous factors possibly playing a role during this developmental period will likely be of interest to researchers in the field, we feel it is beyond the scope of this study to explore these factors in detail, deferring to future research using an experimental protocol designed specifically to do so.

Finally, our findings show that although VPT infants exhibit increasing BOLD variability and progressive cortical microstructural maturation from preterm birth to TEA, by TEA they nonetheless demonstrate reduced functional and microstructural cortical maturation compared with FT newborns, reinforcing VPT infants' distinct developmental trajectory.

Reviewer #2:

It is very interesting to study the difference between VPT and FT from the perspective of BOLD signal variability and its relationship with cortical microstructure and gene expression. Addressing the interplay between BOLD variability from resting-state fMRI, microstructure measures from diffusion MRI, and gene expression is a major strength of this work. However, there are notable concerns about the fundamental mechanisms of BOLD variability and rigorousness of methodology, which limit the impact and robustness of the current work.

Major Comments

- 1. The fundamental mechanisms underlying BOLD variability are not clear. Thus, it does not appear robust to establish a study centered on a measure with its mechanism not well understood. The limited citations provided to explain BOLD variability are constrained to few groups, raising the question if BOLD variability is broadly reproducible or widely applicable.**

We thank the reviewer for their comment, and we agree that the fundamental mechanisms underlying BOLD variability are not yet completely understood. However, converging evidence from multiple independent research groups supports its interpretive value as a proxy for spontaneous neural dynamics and network-level organization. We have revised the introduction to provide additional background and include further relevant references (changes, in yellow, can be seen below and in the manuscript).

Several independent groups have investigated BOLD variability across different populations and age ranges using complementary methodological approaches. In fact, BOLD variability has been operationalized using slightly different metrics, including standard deviation (SD), mean squared successive difference (MSSD), as well as the closely related amplitude of low-frequency fluctuations (ALFF/fALFF). Although ALFF focuses on a specific low-frequency band (0.01–0.08 Hz), whereas SD and MSSD summarize overall temporal variance, the dominant power of the BOLD signal lies within the low-frequency range (0.01–0.1Hz). All these metrics are mathematically related and empirically correlated. Zhang et al. (2022) and Nomi et al. (2017) demonstrated that SD and MSSD are highly correlated in resting-state fMRI data, and ALFF and SD approaches have been recognized as capturing similar aspects of BOLD signal fluctuations over time (Lim et al., 2021; Millar et al., 2020). Therefore, these metrics capture very similar spontaneous BOLD fluctuations, which reflect underlying neural dynamics. Findings from multiple research groups demonstrate that BOLD variability is a widely applicable and reproducible measure.

We present a table below summarizing the metrics used by different groups and the populations studied. The limited number of studies particularly in developmental populations, highlights the need for more research during this period. Our study further extends its application to very preterm infants before term-equivalent age (prior to the typical time of birth), a population that has not been studied using these metrics. We have included additional references from various groups in the introduction and discussion, when relevant, as shown below and in the manuscript.

Introduction

However, the “BOLD signal variability,” characterized by its variance, captures the magnitude of moment-to-moment regional variations in BOLD signal (Garrett et al., 2010, 2011, 2013). It is thought to indirectly reflect spontaneous brain activity arising from moment-to-moment fluctuations in neuronal activity. In fact, BOLD variability has been shown to correlate with electrophysiological measures of neural dynamics, including EEG power and temporal signal variability (Baracchini et al., 2021; McIntosh et al., 2010; Misisic et al., 2011; Uddin, 2020; Vakorin et al., 2011). These temporal fluctuations in neuronal activity, as measured by electrophysiological and EEG studies, are thought to contribute to synaptic connectivity and associate with better cognitive performance (Faisal et al., 2008; McIntosh et al., 2008; Misisic et al., 2011; Uddin, 2020; Vakorin et al., 2011), thereby underlying brain function.

Although BOLD variability is affected by vascular factors such as baseline cerebral blood flow and neurovascular coupling (Whittaker et al., 2016), studies combining BOLD and arterial spin labeling (ASL) imaging show that group differences remain even after accounting for these vascular influences (Garrett et al., 2017; Roberts R.P., 2023). This evidence indicates that BOLD variability reflects not only vascular effects but also neural dynamics, supporting its use as a meaningful index of underlying functional brain activity.

In adults, BOLD variability varies across cortical regions, with greater variability in association and transmodal regions (Baracchini et al., 2021). Greater BOLD

variability has been linked to increased network organization, greater cognitive performance and increased task difficulty across different studies (Baracchini et al., 2021; Baracchini et al., 2023; Garrett et al., 2011, 2013; Goodman et al., 2024; Nomi et al., 2017; Protzner et al., 2013; Roberts R.P., 2023; Zhang et al., 2022), highlighting its functional relevance.

BOLD variability typically increases during childhood (Wang et al., 2021) and serves as a strong predictor of age across the lifespan (Garrett et al., 2010), then declining with aging (Baracchini et al., 2023; Garrett et al., 2013; Grady & Garrett, 2014; Rieck et al., 2022). To date, no studies have examined changes in BOLD variability in infants prior to term-equivalent age (TEA).

Discussion, Section “Preterm birth impacts BOLD variability and cortical microstructure at TEA”:

Compared to FT newborns, very premature infants at TEA showed reduced BOLD-variability specifically in sensory RSNs (auditory and visual), PCC, thalamus and limbic, denoting the impact of preterm birth on cortical functional maturation. These results align with previous work showing that BOLD variability, measured as ALFF (amplitude of low frequency fluctuations), is reduced in moderate-to-late preterm infants at TEA, compared to FT newborns, predominantly in the primary sensory and motor cortices, and in the posterior cingulate cortex and precuneus (Wu et al., 2016).

Table R1. Non-exhaustive summary table of metrics used across groups (with full citations at the end):

Metric	Study Group	Population
BOLD SD	(Garrett et al., 2010, 2011, 2013; Garrett et al., 2017)	Adults & Older adults
	(Roberts R.P., 2023)	Adults
	(Goodman et al., 2024)	Lifespan
	(Protzner et al., 2013)	Adults
	(Millar et al., 2020)	Adults and older adults
	(Lim et al., 2021)	Adults
	(Scarapicchia et al., 2018)	Older adults
MSSD (Mean Squared Successive Difference)	(Zhang et al., 2022) MSSD & BOLD SD	Young adults
	(Nomi et al., 2017) MSSD & BOLD SD	Lifespan
	(Baracchini et al., 2021; Baracchini et al., 2023)	Adults
	(Easson & McIntosh, 2019)	Children and adolescents
	(Boylan et al., 2021)	Adults and older adults
ALFF / fALFF	(Wu et al., 2016)	Moderate-to-late preterm infants
	(Huang et al., 2020)	Neonates vs adults

	(Zang et al., 2007) (Zou et al., 2008) (Zhao et al., 2018)	Children Children and adults Young adults
	(Long et al., 2017)	Children
	(Montala-Flaquer et al., 2022)	Adults and older adults
	(Lee & Hsieh, 2017)	Adults
	(Zheng et al., 2019)	Older adults

- 2. Methodology needs to be further improved to be more rigorous. A) For example, the description of the LASSO model is relatively vague. It is not clear how the regression was performed, how significance was determined, or what the model performance was. In addition to listing the selected three measures, the manuscript should report the actual sample size, model fit statistics, and p-values for all predictors. B) As another example, the analysis of cortical gray matter microstructure using diffusion MRI should consider mitigating partial volume effects.**

We thank the reviewer for their comment. We have revised overall the Results section, adding additional details where necessary, as highlighted in the manuscript.

A)

In particular, we have now clarified the LASSO analysis in the Methods section, as described below in yellow and in the manuscript.

In resume, we performed the LASSO regression on the group-average 9 region \times 10 metric matrix ($n = 9$ regions) to identify the microstructural measures most strongly associated with regional changes in BOLD variability. This region-level approach was chosen to emphasize biologically interpretable spatial patterns, smoothing out inter-subject variability and noise. Subject-level longitudinal analyses would require accounting for the non-independence of multiple regions within the same subject. Since regions from the same subject are correlated (not independent), performing LASSO at the subject level would violate regression assumptions, which is why we opted for the group-average approach.

Regarding significance testing, LASSO is a penalized regression and feature-selection method. Therefore, classical p-values for individual coefficients are not defined or appropriate, as penalization biases coefficient estimates and violates standard inference assumptions. Instead, variable importance is determined by whether the predictors retain non-zero coefficients at the cross-validated λ (the LASSO penalty parameter), rather than by hypothesis testing. In this study, λ was selected using cross-validation, and model generalizability was explicitly evaluated using leave-one-region-out cross-validation. Reported regression coefficients therefore reflect relative feature importance, rather than statistically independent effects. This is consistent with the statistical literature and with standard practice.

Model performance statistics are reported in the Results section. We have updated the manuscript to reflect the revised Lasso-based analyses. In the current version, using the original “optimal pure Lasso penalty” ($\alpha = 1, \lambda_1$ se), we now report the root mean square error (RMSE), the optimal regularization parameter (λ) and the cross-validated coefficient of determination (R^2_{100Cv}) calculated using the leave-one-region-out cross-validation, providing a more robust quantitative assessment of model performance compared with the previous version. In addition, we explored less penalized models, including “Lasso Elastic Net” ($\alpha = 0.5, \lambda_{\min}$) and “pure Lasso” with a smaller penalty ($\alpha = 1, \lambda_{\min}$). All changes are highlighted below in yellow and have been added to the manuscript.

We acknowledge that the Lasso-based results have changed slightly following revision, given the implementation of leave-one-region-out cross-validation, as well as an identified error in outlier selection for some metrics prior to computing regional averages. Importantly, these updates do not alter the main findings or overall interpretation. All analyses are fully reproducible using the “Source Data” provided with the manuscript, along with the R code used for all analyses, which will be made publicly available on Zenodo upon publication.

Methods, Section “MRI data analysis”:

Longitudinal (from 33- to 40-wGA) and cross-sectional differences (at TEA, between VPT at 40-wGA and FT newborns) in both cortical BOLD variability and microstructural diffusivities were assessed using paired-samples t-tests and independent samples t-tests, respectively, with FDR correction for multiple comparisons. Prior to statistical testing, outliers were identified for each metric, by region and timepoint, based on $1.5 \times$ interquartile range (IQR). Values falling below $Q1 - 1.5 \times IQR$ or above $Q3 + 1.5 \times IQR$ were excluded from subsequent analyses. All analyses and visualizations were performed using RStudio (R version 4.4.1).

For visualization of inter-individual variability using boxplots, all subject-level data were normalized by the maximum absolute change across all participants and regions for each metric, thereby constraining values to the range $[-1, 1]$ while preserving the directionality of effects. These globally normalized values were used to generate boxplots, enabling comparison of relative regional patterns across participants and across metrics, independent of absolute effect magnitude. Additional boxplots displaying the raw, non-normalized data, are provided in Supplementary Material (Figure S2).

For bar-plot visualizations and the clustering analysis, for each metric, mean regional changes were first computed for each cortical resting-state network region and then normalized per metric by dividing by the largest absolute mean regional change, preserving the directionality of effects. This scaling yielded a group-level region \times metric matrix representing the relative spatial distribution of longitudinal changes across cortical resting-state network regions.

For Lasso-based regression modelling, the same group-level region \times metric matrix was used, but expressed as absolute values, in order to focus on the magnitude of change, irrespective of the direction. This approach prevents potential cancellation of positive and negative effects across regions and ensures that predictors with strong effects in opposite directions are appropriately retained. Microstructural diffusivities were clustered using the R-package and function ConsensusClusterPlus (k-means base algorithm and Euclidean distance) (Wilkerson & Hayes, 2010), to identify stable clusters of regional diffusivities. The

optimal number of clusters was determined using the elbow method, based on the within-cluster sum of squares (WCSS). This approach enabled grouping cortical regions based on the similarity of their longitudinal microstructural maturation, and to compare these microstructural clusters with those derived from BOLD variability longitudinal changes.

Lasso-based sparse multivariate regression models were used to identify the most relevant microstructural predictors of BOLD variability longitudinal changes, between 33- to 40-wGA. Analyses were performed using the glmnet package in R with leave-one-region-out cross-validation. The analysis was conducted on the group average 9 region x 10 metric matrix ($n = 9$ regions), where BOLD variability change served as the dependent variable and nine microstructural metrics (extraMD, extrTrans, diff, intra, MD, FA, MK, AK, RK) served as predictors. All metrics were standardized prior to modelling, as previously described. Model selection was performed using leave-one-region-out cross-validated mean squared error, and for each penalty value, RMSE and R^2_{100Cv} were calculated. Two penalty choices were evaluated: the more conservative penalty corresponding to one standard error above the minimum (λ_1 se) and the less penalized minimum-error solution (λ_{min}). In addition to a pure LASSO model ($\alpha = 1$), a LASSO Elastic Net model ($\alpha = 0.5$) was examined to allow for partial retention of correlated predictors. Predictors with non-zero coefficients at the selected penalty values were considered relevant.

Results, Section “Relationship between longitudinal cortical microstructural changes and BOLD variability delta changes”:

To identify the microstructural metrics that best predicted the observed changes in BOLD variability, we employed Lasso-based sparse multivariate regression models on the regional-average data. Out of the 9 microstructural metrics, 4 features (MD, RK, AK and FA) were retained, at the optimal pure LASSO penalty (λ_1 se = 0.015, $\alpha = 1$), as predictors of regional BOLD variability changes from 33- to 40-wGA. MD showed the strongest association ($\beta = 0.419$), followed by RK ($\beta = 0.402$), AK ($\beta = 0.353$), and FA ($\beta = 0.332$). The model achieved a high training fit ($R^2 = 0.965$); however, leave-one-region-out cross-validation indicated that these longitudinal changes explained 28% of the observed variance in BOLD variability ($R^2_{100Cv} = 0.28$, RMSE = 0.054). Using less penalized models - a Lasso Elastic Net ($\alpha = 0.5$, $\lambda_{min} = 0.0032$) and a pure LASSO ($\lambda_{min} = 0.0041$, $\alpha = 1$) - additional metrics (intra, diff, extraMD, MK) were retained, increasing the explained variance to 61–64% ($R^2_{100Cv} = 0.61$ – 0.64 , RMSE = 0.0295–0.032, respectively).

B)

We thank the reviewer for highlighting this point. We acknowledge that partial volume effects can influence diffusion MRI measures, particularly in cortical gray matter. In our study, we minimized these effects by restricting analyses to cortical regions defined by our RS parcellation atlas and averaging metrics across all voxels within each region, reducing the influence of outlier voxels. While residual partial-volume effects cannot be fully eliminated, this approach is consistent with current best practices in neonatal diffusion MRI studies. Importantly, the pipeline is fully automated, and any systematic partial volume effects would be consistent

across subjects, allowing valid longitudinal comparisons and between-group analyses. We have also highlighted this limitation in the manuscript (section 4).

- 3. The relatively small sample size, combined with the lack of validation on publicly available datasets, raises concerns about the generalizability of the findings. The results would be more convincing with rigorous validation. Although the authors note the difficulty of acquiring longitudinal preterm participants, the Developing Human Connectome Project (dHCP) contains a large cohort of both preterm and term infants, including longitudinal scans. Leveraging this public dataset would greatly strengthen the conclusions.**

We thank the reviewer for raising the point regarding validation on publicly available datasets. We carefully considered the Developing Human Connectome Project (dHCP) data; however, according to the metadata from dHCP-2 and dHCP-3 releases, only 18 preterm subjects have longitudinal scans within the specific $32^{4/7} - 34^{3/7}$ wGA window evaluated in our study. From these, only 9 have a second scan within the TEA evaluated window $38^{5/7} - 41^{1/7}$. From the 9 subjects with data acquired within these developmental windows, **only 8** subjects had both diffusion and RS-fMRI data acquired at both times points. Furthermore, these numbers refer to acquired data before preprocessing. Very likely, not all the available subjects will have both raw diffusion and resting-state fMRI data that would pass quality control and be analyzable.

Given this extremely small number, validation using the dHCP cohort would be statistically underpowered and could not provide a meaningful assessment of generalizability. Importantly, our cohort is highly homogeneous, comprising healthy very preterm infants (born before 32 wGA) without major neonatal complications, scanned within a tightly defined developmental window, which increases the reliability of our findings and reduces potential confounding effects. Therefore, we have chosen not to follow the Reviewer’s suggestion to leverage the available dHCP data. However, we acknowledge that the inclusion of a validation set would strengthen our conclusion and have included a statement to this effect, as highlighted, in **Section “Limitations”**: **Future studies with a larger, longitudinal, age-matched cohort will be important to strengthen the external validity of our findings.**

Table R2. Summary of the dHCP preterm longitudinal data, grouped per scan age and sex.

dHCP data - Preterm infants	N_participants	N_female	N_male
Preterm with longitudinal data	84	41	43
With First scan ($32^{4/7}$ - $34^{3/7}$ wGA)	18	10	8
With Second scan ($38^{5/7}$ - $41^{1/7}$ wGA)	35	17	18
With Both ranges met	9	6	3
Ranges met + both Diffusion and fMRI data acquired	8	6	2

Table R3. Subject-level description of the dHCP preterm longitudinal data within the specific GA window evaluated in our study, including sequence acquisition (MSDI and fMRI) availability.

Subject count	participant_id	session_id	scan_number	scan_age	birth_age	sex	MSDI	fMRI
1	CC00248XX18	83000	1	33	30.2857143	female	yes	yes
	CC00248XX18	92700	2	39.86	30.2857143	female	yes	yes
2	CC00301XX04	96400	1	32.71	28.7142857	female	yes	yes
	CC00301XX04	113001	2	40	28.7142857	female	yes	yes
3	CC00305XX08	98101	1	32.71	31.8571429	female	yes	yes
	CC00305XX08	115700	2	41.14	31.8571429	female	yes	yes
4	CC00525XX14	150600	1	33.57		31 female	yes	yes
	CC00525XX14	165900	2	40.86		31 female	yes	yes
5	CC00617XX15	176500	1	34.14	31.5714286	male	yes	yes
	CC00617XX15	188400	2	40.14	31.5714286	male	yes	yes
6	CC00686XX19	198800	1	33.43	32.5714286	male	yes	yes
	CC00686XX19	209100	2	39.71	32.5714286	male	yes	yes
7	CC00829XX21	17610	1	33.14	32.2857143	female	yes	yes
	CC00829XX21	27411	2	40.29	32.2857143	female	yes	yes
8	CC00838XX22	21910	1	33.14	31.7142857	female	yes	yes
	CC00838XX22	30610	2	40	31.7142857	female	yes	yes
9	CC01208XX12	144731	1	34.29	25.4285714	male	yes	yes
	CC01208XX12	149630	2	40.43	25.4285714	male	yes	no

4. While the authors chose 9 microstructure measures from diffusion MRI, BOLD variability was used as the only measure derived from resting state fMRI. Other relatively more widely adopted measures from resting-state fMRI such as amplitude of low frequency fluctuations (ALFF) or regional homogeneity (ReHo) should be included to test the findings or the authors should justify exclusion of these other more widely used measures from resting state fMRI. Lacking these measures from resting state fMRI weakens the functional analysis and limits the ability to fully address how microstructural and biological development support functional maturation.

We thank the reviewer for their comment. As noted, BOLD variability has been operationalized using slightly different metrics, including standard deviation (SD), as well as the closely related amplitude of low-frequency fluctuations (ALFF/fALFF). Both these measures aim to capture the amplitude of spontaneous moment-to-moment regional BOLD signal oscillations. In contrast, ReHo (Regional homogeneity) measures local functional connectivity - specifically, the synchronous activity of a voxel with its nearest neighbors - and is not a measure of BOLD variability. This local functional synchronous activity (local correlatedness) is measured through Kendall correlation between RS-fMRI BOLD signal at the individual voxel (volume unit) level and its nearest neighbors (local connectivity), which is very different from BOLD signal variability.

Our study aimed to evaluate specifically BOLD variability and its underpinnings. More specifically, how the observed changes in BOLD variability align with regional measures of cortical microstructure, and the patterns of gene expressions that could explain the biological correlates underlying the observed microstructural

and functional changes. For this reason, calculating the ReHo is not within the scope of this work.

However, to address the reviewer's suggestion, we have additionally performed the same longitudinal and cross-sectional analysis using ALFF and fALFF.

Both ALFF/fALFF and BOLD SD capture similar aspects of BOLD signal fluctuation over time (Lim et al., 2021; Millar et al., 2020). While ALFF focuses on a specific low-frequency band (0.01–0.08 Hz), SD summarizes overall temporal variance. fALFF further normalizes the amplitude of spontaneous BOLD fluctuations within the low-frequency band (0.01–0.08 Hz) by the total spectral power across all frequencies, aiming to reduce the influence of noise and nonspecific global signal contributions. However, some studies suggest that this normalization may reduce reliability relative to ALFF (Jia et al., 2020; Sbairhat et al., 2022). Overall, since the dominant power of the BOLD signal lies within the low-frequency range (0.01–0.1 Hz), all these measures are highly comparable. As shown in Figure S1 and S5, the longitudinal and cross-sectional results obtained using ALFF/fALFF are consistent with those from BOLD SD, supporting the validity of our findings.

We chose BOLD variability as our primary measure because it captures moment-to-moment fluctuations across the full frequency spectrum, providing a broad and sensitive index of spontaneous neural dynamics. ALFF and fALFF are restricted to a predefined low-frequency band and are more dependent on preprocessing choices, while BOLD variability offers a more general and assumption-free characterization of spontaneous activity, which may be better suited to capturing the physiological and microstructural changes occurring during late gestation.

As requested, for completeness, we performed parallel analyses with ALFF and fALFF, which yielded consistent results. These results have been referenced in the Results and Methods sections, as highlighted in the manuscript, and have been added to the Supplementary Material.

Results, Section “Longitudinal cortical BOLD variability and diffusion microstructural changes during early preterm brain development”:

Complementary analysis using ALFF and fALFF to estimate BOLD variability yielded consistent longitudinal patterns across the same regions (see Supplementary Figure S1).

Results, Section “Effect of preterm birth on BOLD variability and cortical microstructure at TEA”:

Complementary analysis using ALFF and fALFF to estimate BOLD variability yielded consistent cross-sectional patterns across the same regions (see Supplementary Figure S5).

Methods, Section “Regional cortical BOLD variability estimation”:

In addition, ALFF (Amplitude of Low-Frequency Fluctuations) was computed by transforming each regional time series into the frequency domain using the Fast Fourier Transform to obtain the power spectrum (Feng et al., 2024; Huang et al., 2020; Yan et al., 2021). The square root of the power spectrum within the 0.01–0.08 Hz (low) frequency band was averaged to obtain regional ALFF values.

Regional fALFF (fractional ALFF) was then calculated as the ratio of the low-frequency (0.01–0.08 Hz) ALFF to the total ALFF across all frequencies of the power spectrum. These measures were used in parallel to confirm that the results of BOLD SD analyses of spontaneous BOLD fluctuations were consistent across other different metrics estimating BOLD variability.

We selected BOLD variability (BOLD SD) as the primary measure because it captures moment-to-moment fluctuations across the full frequency spectrum, providing a broad and sensitive index of spontaneous neural dynamics. In contrast, ALFF and fALFF are restricted to a predefined low-frequency band and are more dependent on preprocessing choices, whereas BOLD variability provides a more general, assumption-free characterization of spontaneous activity, which is better suited to capturing physiological and microstructural changes during late gestation. ALFF and fALFF results are presented in Supplementary Material.

BOLD variability, ALFF and fALFF Longitudinal Changes per Region

Figure S1. Longitudinal BOLD, ALFF and fALFF changes from 33 to 40 weeks. Bar plots illustrate the group-averaged BOLD variability, ALFF and fALFF longitudinal changes in VPT infants from 33 wGA to TEA across networks, with each metric scaled to its own largest absolute mean regional change. Embedded boxplots show the distribution of individual subject changes within each RSN for each metric, with individual values scaled to the largest absolute subject-level change observed across all RSNs for that metric. Significant changes are indicated by asterisks (“*”p<0.05, “**”p<0.01, “***”p<0.001, after FDR correction). PCUN= precuneus, PCC = posterior cingulate cortex, SSM = sensorimotor, VIS = visual, AUD = auditory, THAL = thalamus, PFC = prefrontal cortex.

Figure S5. BOLD variability (a), ALFF (b) and fALFF (c) differences, between FT and VPT infants at TEA. Boxplots represent the distribution of individual subject values (FT in blue, VPT in coral) within each network, illustrating inter-subject variability. Bar plots illustrate the average group difference (FT-VPT) for each network. Significant changes are indicated by asterisks (“*” $p < 0.05$, “**” $p < 0.01$, “***” $p < 0.001$, after FDR correction). PCUN= precuneus, PCC = posterior cingulate cortex, SSM = sensorimotor, VIS = visual, AUD = auditory, THAL = thalamus, PFC = prefrontal cortex.

Minor Comments:

5. While the Introduction provides useful background on diffusion models, it does not sufficiently justify why DTI measures (e.g., FA, MD) were applied to study cortical microstructure, given their traditional use in white matter.

We thank the reviewer for their comment. Although DTI has traditionally been applied to WM, it has also been applied and successfully used to study GM. As noted in the introduction, DTI does not hold intrinsic diffusivity assumptions, capturing variations in diffusion patterns, reason why it is one of the diffusion models widely accepted to be used both in WM and GM: “Traditional signal representation models of diffusion, such as Diffusion Tensor Imaging (DTI), provide insights into GM organization by measuring the anisotropic diffusion of water molecules within the tissue, without intrinsic diffusivity assumptions, capturing variations in diffusion patterns (Le Bihan et al., 2001)”.

Importantly, several neonatal and fetal MRI studies have demonstrated that DTI-derived measures are sensitive to early cortical microstructural maturation. In fact, it has been very useful to detect and to state cortical complexification occurring before term age, during the third trimester of pregnancy. During this period there

is an important increase in dendritic arborization, and disappearance of the radial glia, leading to a more complex multidirectional and less radial diffusion environment (Bystron et al., 2008; Cameron & Rakic, 1991; He et al., 2020; Marin-Padilla, 1992; Rakic, 2003), which leads to the well described decrease in cortical FA, observed when using DTI to study early cortical maturation (Ball et al., 2013; Bataille et al., 2019; Eaton-Rosen et al., 2015; McKinstry et al., 2002). For this reason, we included DTI in our analysis as a standard baseline diffusion model, allowing us to build on a well-established literature on cortical DTI findings in the developing brain, and we complemented it with the more advanced SMT and DKI models.

- 6. Since the authors used three different diffusion models, it would be beneficial to specify the b-values used in each diffusion model in the Methods.**

Thank you for the suggestion. All diffusion models (SMT, DKI and DTI) were estimated using the same multi-shell acquisition. To be more explicit, we have added this to **Methods, Section “Diffusion cortical microstructural maps”**: After pre-processing, MSDI data was analyzed using three different methodological approaches, each applied to the same set of b-values (b=200, b=1000, b=2000 s/mm²).

In the same section, we also state: “For both DTI and DKI metrics, we used the code publicly available from DIPY library (https://docs.dipy.org/stable/interfaces/reconstruction_flow.html), applying the DKI model to the full MSDI data.” This approach avoids inconsistencies that may arise when DTI is fit only to a low-b subset and improves tensor stability by incorporating the additional high-b information.

- 7. The definition of “clusters” is not clear. In BOLD variability, “clusters” appear to denote groups of regions showing significant increases between two timepoints. But in diffusion MRI, k-means clustering (k=3) was used without justification of why three clusters were chosen. To avoid confusion, the authors could consider using “groups” for the BOLD variability groups of regions and provide validation (e.g., elbow method, silhouette score) for the k-means choice of clusters.**

We thank the reviewer for their suggestion. We have changed the definition of “clusters” to “groups” when referring to the regions with or without significant BOLD variability changes.

In addition to this grouping based on BOLD variability changes, we wanted to assess if the longitudinal microstructural maturational changes would result in a similar regional pattern grouping (independently of BOLD variability changes). For that reason, we have conducted an additional clustering analysis, as described in **Methods, Section “MRI data analysis”** and **Supplementary Material Figure S3.**: “Microstructural diffusivities were clustered using the R-package and function *ConsensusClusterPlus* (k-means base algorithm and Euclidean distance) (Wilkerson & Hayes, 2010), to identify stable clusters of regional diffusivities. The optimal number of clusters was determined using the elbow method, based on the within-cluster sum of squares (WCSS).” We selected k = 3 as the optimal number

of clusters using the elbow method, which showed a clear inflection point at this value. Beyond $k = 3$, the reduction in within-cluster sum of squares (WCSS) diminished substantially, indicating limited gain in compactness with additional clusters. We have added Figure S2 a), which displays the elbow method.

We acknowledge that the consensus clustering analysis results have changed slightly following revision. This is because we now used the group-level region \times metric matrix (representing the relative spatial distribution of longitudinal changes across cortical resting-state network regions) preserving the directionality of effects, as explained in Methods section. For clustering, retaining the sign of metric changes is important, as positive versus negative changes reflect distinct patterns of microstructural maturation across regions. This contrasts with Lasso-based regression, which focuses on the magnitude of change, and for which the directionality of effects is not informative for feature selection. Importantly, these updates do not alter the main findings or overall interpretation.

As explained in Figure S2 paragraph: “Three distinct regional microstructural maturational patterns were identified, aligning with the central-to-peripheral and posterior-to-anterior known gradients in brain maturation, in agreement with the brain myelination order of Kinney (Kinney et al., 1988). Microstructural cluster 2 aligns with BOLD SD group 1, while Microstructural cluster 3 is more similar to BOLD SD group 2. The thalamus emerged as a separate cluster, reflecting its distinct microstructural profile.” This microstructural spatial distribution, although not perfectly identical, is similar to the regional patterns identified by BOLD variability changes, highlighting that the primary sensory-motor and proto-DMN regions exhibit both functional and microstructural changes distinct from those of other brain regions”.

Figure S2. Results from the consensus clustering of the combined 9 distinct

a) Elbow method showing an inflection point at $K = 3$, suggesting an optimal clustering solution with three clusters. b) Heatmap depicting the regional maturational patterns from 33- to 40-wGA, with rows and columns representing the RSNs and color intensity in each cell reflecting the composite changes over time of the microstructural metrics. c) Composition of the identified 3 regional clusters of microstructural delta changes (Cluster 1: Thalamus, Cluster 2: PCC, Visual, PCUN, SSM, Auditory, Limbic; Cluster 3: PFC, Paralimbic). d) Brain plots illustrating the 3 distinct regional clusters depicted by the consensus clustering analysis, color-coded according to regional cluster (red = cluster 1, orange = cluster 2, blue = cluster 3).

8. The authors might want to be cautious about using “trajectory”, as only two timepoints were included in the manuscript.

We thank the reviewer for their suggestion and have removed the word “trajectory” from the discussion.

9. There is a typo in the last sentence of the Introduction: “weather” should be corrected to “whether.”

We thank the reviewer for detecting this typo and we have corrected it accordingly.

Reviewer #3 (Remarks to the Author):

Thank you for asking me to review this interesting paper examining neonatal brain development. The work combines MRI data with open access transcriptomic data to produce interesting and plausible biologic insights. The combination of BOLD as a measure of cortical function with transcription data is as far as I know completely novel and of great interest. The combination of MRI with transcription data is an emerging and promising area of science and Ball is the world leader in developing this approach in the developing brain. The paper leverages longitudinal imaging data encoding the 'experiment of nature' that is preterm delivery. Premature birth is a wide-ranging and robust disrupter of brain development, which has been shown (by these authors as well as many others) to affect almost every aspect of brain development and so provides a useful 'positive control' reality check on these results.

1. The imaging data appropriately collected with reasonably up to date sequences. The cohort is small, and indeed quite a lot smaller than most substantial studies currently being published. The data were collected by the researchers, and they probably trust their own data quality, relying on the longitudinal data to add value to the small cohort. Nevertheless, in the same way that they used open access transcriptomic data, they could have replicated their results using the open access imaging data now freely available, which are an order of magnitude larger than the current dataset, and which some of this group have used in the past.

We thank the reviewer for their relevant comment. We carefully chose a longitudinal design for our study because both BOLD-variability and microstructural diffusion metrics are subject-dependent, and single time-point cross-sectional group comparisons are less reliable for capturing developmental changes. Longitudinal within-subject data, collected at well-defined time-points, allows us to more robustly track individual trajectories of cortical maturation, which is particularly critical during this period of rapid developmental changes.

As noted in response to Reviewer 2 above, we have considered the Developing Human Connectome Project (dHCP) data; however, according to the metadata from dHCP-2 and dHCP-3 releases, only 18 preterm subjects have longitudinal scans within the specific $32^{4/7} - 34^{3/7}$ wGA window evaluated in our study. From these, only 9 have a second scan within the TEA evaluated window $38^{5/7} - 41^{1/7}$. From the 9 subjects with data acquired within these developmental windows, **only 8** subjects had both diffusion and RS-fMRI data acquired at both times points. Furthermore, these numbers refer to acquired data before preprocessing. Very likely, not all the available subjects will have both raw diffusion and resting-state fMRI data that would pass quality control and be analyzable. Given this extremely small number, validation using the dHCP cohort would be statistically underpowered and could not provide a meaningful assessment of generalizability. Importantly, our cohort is highly homogeneous, comprising healthy very preterm infants (born before 32 wGA) without major neonatal complications, scanned within a tightly defined developmental window, which increases the reliability of our findings and reduces potential confounding effects.

Therefore, we have chosen not to leverage the available dHCP data. However, we acknowledge that the inclusion of a validation set would strengthen our conclusion and have included a statement to this effect, as highlighted, in **section “Limitations”**: **Future studies with a larger, longitudinal, age-matched cohort will be important to strengthen the external validity of our findings.**

Table R2. Summary of the dHCP preterm longitudinal data, grouped per scan age and sex.

dHCP data - Preterm infants	N_participants	N_female	N_male
Preterm with longitudinal data	84	41	43

With First scan (32 ^{4/7} -34 ^{3/7} wGA)	18	10	8
With Second scan (38 ^{5/7} -41 ^{1/7} wGA)	35	17	18
With Both ranges met	9	6	3
Ranges met + both Diffusion and fMRI data acquired	8	6	2

Table R3. Subject-level description of the dHCP preterm longitudinal data within the specific GA window evaluated in our study, including sequence acquisition (MSDI and fMRI) availability.

Subject count	participant_id	session_id	scan_numbe	scan_age	birth_age	sex	MSDI	fMRI
1	CC00248XX18	83000	1	33	30.2857143	female	yes	yes
	CC00248XX18	92700	2	39.86	30.2857143	female	yes	yes
2	CC00301XX04	96400	1	32.71	28.7142857	female	yes	yes
	CC00301XX04	113001	2	40	28.7142857	female	yes	yes
3	CC00305XX08	98101	1	32.71	31.8571429	female	yes	yes
	CC00305XX08	115700	2	41.14	31.8571429	female	yes	yes
4	CC00525XX14	150600	1	33.57		31 female	yes	yes
	CC00525XX14	165900	2	40.86		31 female	yes	yes
5	CC00617XX15	176500	1	34.14	31.5714286	male	yes	yes
	CC00617XX15	188400	2	40.14	31.5714286	male	yes	yes
6	CC00686XX19	198800	1	33.43	32.5714286	male	yes	yes
	CC00686XX19	209100	2	39.71	32.5714286	male	yes	yes
7	CC00829XX21	17610	1	33.14	32.2857143	female	yes	yes
	CC00829XX21	27411	2	40.29	32.2857143	female	yes	yes
8	CC00838XX22	21910	1	33.14	31.7142857	female	yes	yes
	CC00838XX22	30610	2	40	31.7142857	female	yes	yes
9	CC01208XX12	144731	1	34.29	25.4285714	male	yes	yes
	CC01208XX12	149630	2	40.43	25.4285714	male	yes	no

2. The image analysis is reasonable. It appears that there was no motion correction applied to the scan data, but a reasonable cut-off for exclusion of scans was used to remove corrupted datasets. It is a bit controversial whether this scrubbing of individual data (which disrupts the time course) affects fMRI interpretation, but my view is that it is unlikely to invalidate measures of BOLD variability. Variability is a nice and under-explored measure of cortical function, the results are interesting and as far as I know novel, and the functional locations revealed are interesting. Spherical means is a good method for analysing diffusion data and the results are consistent with previous studies using different approaches.

We thank the reviewer for this comment. The measure of BOLD variability used in our study reflects the overall amplitude of signal fluctuations (standard deviation across time) and does not rely on the temporal sequence or frequency characteristics of the signal. Therefore, removing high-motion volumes (“scrubbing”) does not bias or distort the amplitude measure, as it would only minimally affect the overall variance calculation. By contrast, metrics such as the Hurst exponent (HE) or the mean squared successive difference (MSSD) depend on temporal correlations or the temporal ordering of time points. Scrubbing disrupts the time series, which can strongly influence these measures. In this

sense, BOLD variability is inherently more robust to motion-related data removal, supporting the reliability of our results despite excluding corrupted scans.

- 3. The imaging results are plausible and interesting, with the increases in visual and sensory-motor cortex in late gestation as expected given known developmental pathways, and to is also interesting that they did not find BOLD variability increases in pre-frontal cortex or limbic- it is plausible to interpret this as the frontal cortex not having started its rapid development while the limbic system may have completed its comparable phase. However, it would have been interesting to see more data on temporal cortex which is rapidly developing at this time.**

We thank the reviewer, and we agree with these important insights. Our resting-state networks (RSNs) atlas was generated from the combination of RSNs identified in our cohort of infants, including both VPT at 33- and 40-wGA, as well as the FT infants, combined into a cortical parcellation atlas. We chose this approach since we preferred to use RSN directly generated from our data during this developmental time, rather than applying an external atlas to our data. Consequently, our analysis is constrained to the RSN that were observed. Notably, we observed an increase in BOLD variability in the auditory network from 33 to 40 weeks' GA. The temporal pole was included within the paralimbic belt network, together with the orbito-frontal cortex, and this network did not show such an increase in BOLD variability. This further supports a different maturational pattern between primary vs paralimbic/associative regions.

- 4. The transcriptomic data and analysis are high quality and appropriate. it is helpful to have the three time points in the data even if there are no imaging data to compare to the first period. The gene set analysis is uncontroversial and Lasso is a reasonable approach to this underdetermined dataset. As with the imaging data the results are plausible and it is re-assuring that neuronal ensheathment is most significant relation, which gives confidence in other pathways detected. It is a shame that the analysis does not make use of the remarkable microbrain histologic parcellation that one of the authors has recently published but relies on small patch location for the transcriptomic analysis, but that does not invalidate the results. The gene sets exposed in the analysis are interesting and add to existing knowledge.**

We thank the reviewer very much for their positive feedback. As discussed above in response to Reviewer 1, despite a growing catalogue of prenatal gene expression studies, including those that span the whole of gestation (e.g.: BrainSpan), or those with multiple cortical regions (e.g. μ Brain), none yet exist that offer both spatial and temporal resolution during this time period. As our window of observation was focused on the third trimester, we chose to leverage the temporal resolution of BrainSpan to examine trends in gene expression over the prenatal window. The cortical regions sampled in the BrainSpan data also cover major regions of the neocortex, allowing for a regional analysis over time. The μ Brain atlas, while offering more detailed spatial resolution, is based on microarray

data from a small window in mid-gestation (15-20weeks), right at the onset of gliogenesis, and, given the rapid structural, cellular and morphological changes during mid- to- late gestation, it is not certain how well this would map onto the imaging observations acquired toward the end of gestation. We anticipate that, with new resources, exploiting both spatial and temporal dimensions will yield interesting future research directions in this area, including the structural and functional maturation of the cortex during gestation.

References

- Ball, G., Srinivasan, L., Aljabar, P., Counsell, S. J., Durighel, G., Hajnal, J. V., Rutherford, M. A., & Edwards, A. D. (2013). Development of cortical microstructure in the preterm human brain. *Proceedings of the National Academy of Sciences of the United States of America*, *110*(23), 9541-9546. <https://doi.org/10.1073/pnas.1301652110>
- Baracchini, G., Masic, B., Setton, R., Mwilambwe-Tshilobo, L., Girn, M., Nomi, J. S., Uddin, L. Q., Turner, G. R., & Spreng, R. N. (2021). Inter-regional BOLD signal variability is an organizational feature of functional brain networks. *NeuroImage*, *237*. <https://doi.org/ARTN118149>
10.1016/j.neuroimage.2021.118149
- Baracchini, G., Zhou, Y., da Silva Castanheira, J., Hansen, J. Y., Rieck, J., Turner, G. R., Grady, C. L., Masic, B., Nomi, J., Uddin, L. Q., & Spreng, R. N. (2023). The biological role of local and global fMRI BOLD signal variability in human brain organization. *bioRxiv*. <https://doi.org/10.1101/2023.10.22.563476>
- Batalle, D., O'Muircheartaigh, J., Makropoulos, A., Kelly, C. J., Dimitrova, R., Hughes, E. J., Hajnal, J. V., Zhang, H., Alexander, D. C., Edwards, A. D., & Counsell, S. J. (2019). Different patterns of cortical maturation before and after 38 weeks gestational age demonstrated by diffusion MRI in vivo. *NeuroImage*, *185*, 764-775. <https://doi.org/10.1016/j.neuroimage.2018.05.046>
- Boylan, M. A., Foster, C. M., Pongpipat, E. E., Webb, C. E., Rodrigue, K. M., & Kennedy, K. M. (2021). Greater BOLD Variability is Associated With Poorer Cognitive Function in an Adult Lifespan Sample. *Cereb Cortex*, *31*(1), 562-574. <https://doi.org/10.1093/cercor/bhaa243>
- Bystron, I., Blakemore, C., & Rakic, P. (2008). Development of the human cerebral cortex: Boulder Committee revisited. *Nat Rev Neurosci*, *9*(2), 110-122. <https://doi.org/10.1038/nrn2252>
- Cameron, R. S., & Rakic, P. (1991). Glial-Cell Lineage in the Cerebral-Cortex - a Review and Synthesis. *Glia*, *4*(2), 124-137. <https://doi.org/DOI10.1002/glia.440040204>
- Drayne, J. P., Mella, A. E., McLean, M. M., Ufkes, S., Chau, V., Guo, T., Branson, H. M., Kelly, E., Miller, S. P., Grunau, R. E., & Weber, A. M. (2024). Long-range temporal correlation development in resting-state fMRI signal in preterm infants: Scanned shortly after birth and at term-equivalent age. *PLOS Complex Syst*, *1*(4), Article e0000024. <https://doi.org/https://doi.org/10.1371/journal.pcsy.0000024>

- Easson, A. K., & McIntosh, A. R. (2019). BOLD signal variability and complexity in children and adolescents with and without autism spectrum disorder. *Dev Cogn Neurosci*, 36, 100630. <https://doi.org/10.1016/j.dcn.2019.100630>
- Eaton-Rosen, Z., Melbourne, A., Orasanu, E., Cardoso, M. J., Modat, M., Bainbridge, A., Kendall, G. S., Robertson, N. J., Marlow, N., & Ourselin, S. (2015). Longitudinal measurement of the developing grey matter in preterm subjects using multi-modal MRI. *NeuroImage*, 111, 580-589. <https://doi.org/10.1016/j.neuroimage.2015.02.010>
- Faisal, A. A., Selen, L. P., & Wolpert, D. M. (2008). Noise in the nervous system. *Nat Rev Neurosci*, 9(4), 292-303. <https://doi.org/10.1038/nrn2258>
- Garrett, D. D., Kovacevic, N., McIntosh, A. R., & Grady, C. L. (2010). Blood oxygen level-dependent signal variability is more than just noise. *J Neurosci*, 30(14), 4914-4921. <https://doi.org/10.1523/JNEUROSCI.5166-09.2010>
- Garrett, D. D., Kovacevic, N., McIntosh, A. R., & Grady, C. L. (2011). The importance of being variable. *J Neurosci*, 31(12), 4496-4503. <https://doi.org/10.1523/JNEUROSCI.5641-10.2011>
- Garrett, D. D., Kovacevic, N., McIntosh, A. R., & Grady, C. L. (2013). The Modulation of BOLD Variability between Cognitive States Varies by Age and Processing Speed. *Cerebral Cortex*, 23(3), 684-693. <https://doi.org/10.1093/cercor/bhs055>
- Garrett, D. D., Lindenberger, U., Hoge, R. D., & Gauthier, C. J. (2017). Age differences in brain signal variability are robust to multiple vascular controls. *Sci Rep*, 7(1), 10149. <https://doi.org/10.1038/s41598-017-09752-7>
- Goodman, Z. T., Nomi, J. S., Kornfeld, S., Bolt, T., Saumure, R. A., Romero, C., Bainter, S. A., & Uddin, L. Q. (2024). Brain signal variability and executive functions across the life span. *Netw Neurosci*, 8(1), 226-240. https://doi.org/10.1162/netn_a_00347
- Grady, C. L., & Garrett, D. D. (2014). Understanding variability in the BOLD signal and why it matters for aging. *Brain Imaging and Behavior*, 8(2), 274-283. <https://doi.org/10.1007/s11682-013-9253-0>
- He, L. X., Wan, L., Xiang, W., Li, J. M., Pan, A. H., & Lu, D. H. (2020). Synaptic development of layer V pyramidal neurons in the prenatal human prefrontal neocortex: a NeuroLucida-aided Golgi study. *Neural Regeneration Research*, 15(8), 1490-1495. <https://doi.org/10.4103/1673-5374.274345>
- Huang, Z., Wang, Q., Zhou, S., Tang, C., Yi, F., & Nie, J. (2020). Exploring functional brain activity in neonates: A resting-state fMRI study. *Dev Cogn Neurosci*, 45, 100850. <https://doi.org/10.1016/j.dcn.2020.100850>
- Jia, X. Z., Sun, J. W., Ji, G. J., Liao, W., Lv, Y. T., Wang, J., Wang, Z., Zhang, H., Liu, D. Q., & Zang, Y. F. (2020). Percent amplitude of fluctuation: A simple measure for resting-state fMRI signal at single voxel level. *PLoS ONE*, 15(1), e0227021. <https://doi.org/10.1371/journal.pone.0227021>
- Kang, H. J., Kawasawa, Y. I., Cheng, F., Zhu, Y., Xu, X., Li, M., Sousa, A. M., Pletikos, M., Meyer, K. A., Sedmak, G., Guennel, T., Shin, Y., Johnson, M. B., Krsnik, Z., Mayer, S., Fertuzinhos, S., Umlauf, S., Lisgo, S. N., Vortmeyer, A.,...Sestan, N. (2011). Spatio-temporal transcriptome of the human brain. *Nature*, 478(7370), 483-489. <https://doi.org/10.1038/nature10523>
- Kinney, H. C., Brody, B. A., Kloman, A. S., & Gilles, F. H. (1988). Sequence of central nervous system myelination in human infancy. II. Patterns of myelination in

- autopsied infants. *J Neuropathol Exp Neurol*, 47(3), 217-234. <https://www.ncbi.nlm.nih.gov/pubmed/3367155>
- Le Bihan, D., Mangin, J. F., Poupon, C., Clark, C. A., Pappata, S., Molko, N., & Chabriat, H. (2001). Diffusion tensor imaging: Concepts and applications. *Journal of Magnetic Resonance Imaging*, 13(4), 534-546. <https://doi.org/DOI 10.1002/jmri.1076>
- Lee, H. H., & Hsieh, S. (2017). Resting-State fMRI Associated with Stop-Signal Task Performance in Healthy Middle-Aged and Elderly People. *Front Psychol*, 8, 766. <https://doi.org/10.3389/fpsyg.2017.00766>
- Lim, M., Jassar, H., Kim, D. J., Nascimento, T. D., & DaSilva, A. F. (2021). Differential alteration of fMRI signal variability in the ascending trigeminal somatosensory and pain modulatory pathways in migraine. *J Headache Pain*, 22(1), 4. <https://doi.org/10.1186/s10194-020-01210-6>
- Long, X., Benischek, A., Dewey, D., & Lebel, C. (2017). Age-related functional brain changes in young children. *NeuroImage*, 155, 322-330. <https://doi.org/10.1016/j.neuroimage.2017.04.059>
- Marin-Padilla, M. (1992). Ontogenesis of the pyramidal cell of the mammalian neocortex and developmental cytoarchitectonics: a unifying theory. *J Comp Neurol*, 321(2), 223-240. <https://doi.org/10.1002/cne.903210205>
- McIntosh, A. R., Kovacevic, N., & Itier, R. J. (2008). Increased brain signal variability accompanies lower behavioral variability in development. *Plos Computational Biology*, 4(7), e1000106. <https://doi.org/10.1371/journal.pcbi.1000106>
- McIntosh, A. R., Kovacevic, N., Lippe, S., Garrett, D., Grady, C., & Jirsa, V. (2010). The development of a noisy brain. *Arch Ital Biol*, 148(3), 323-337. <https://www.ncbi.nlm.nih.gov/pubmed/21175017>
- McKinstry, R. C., Mathur, A., Miller, J. H., Ozcan, A., Snyder, A. Z., Schefft, G. L., Almlı, C. R., Shiran, S. I., Conturo, T. E., & Neil, J. J. (2002). Radial organization of developing preterm human cerebral cortex revealed by non-invasive water diffusion anisotropy MRI. *Cerebral Cortex*, 12(12), 1237-1243. <https://doi.org/DOI 10.1093/cercor/12.12.1237>
- Mella, A. E., Vanderwal, T., Miller, S. P., & Weber, A. M. (2024). Temporal complexity of the BOLD-signal in preterm versus term infants. *Cereb Cortex*, 34(10). <https://doi.org/10.1093/cercor/bhae426>
- Millar, P. R., Petersen, S. E., Ances, B. M., Gordon, B. A., Benzinger, T. L. S., Morris, J. C., & Balota, D. A. (2020). Evaluating the Sensitivity of Resting-State BOLD Variability to Age and Cognition after Controlling for Motion and Cardiovascular Influences: A Network-Based Approach. *Cereb Cortex*, 30(11), 5686-5701. <https://doi.org/10.1093/cercor/bhaa138>
- Miller, D. J., Duka, T., Stimpson, C. D., Schapiro, S. J., Baze, W. B., McArthur, M. J., Fobbs, A. J., Sousa, A. M., Sestan, N., Wildman, D. E., Lipovich, L., Kuzawa, C. W., Hof, P. R., & Sherwood, C. C. (2012). Prolonged myelination in human neocortical evolution. *Proc Natl Acad Sci U S A*, 109(41), 16480-16485. <https://doi.org/10.1073/pnas.1117943109>
- Misic, B., Vakorin, V. A., Paus, T., & McIntosh, A. R. (2011). Functional embedding predicts the variability of neural activity. *Front Syst Neurosci*, 5, 90. <https://doi.org/10.3389/fnsys.2011.00090>
- Montala-Flaquer, M., Canete-Masse, C., Vaque-Alcazar, L., Bartres-Faz, D., Pero-Cebollero, M., & Guardia-Olmos, J. (2022). Spontaneous brain activity in healthy aging: An overview through fluctuations and regional homogeneity.

- Frontiers in Aging Neuroscience*, 14, 1002811. <https://doi.org/10.3389/fnagi.2022.1002811>
- Nomi, J. S., Bolt, T. S., Ezie, C. E. C., Uddin, L. Q., & Heller, A. S. (2017). Moment-to-Moment BOLD Signal Variability Reflects Regional Changes in Neural Flexibility across the Lifespan. *J Neurosci*, 37(22), 5539-5548. <https://doi.org/10.1523/JNEUROSCI.3408-16.2017>
- Protzner, A. B., Kovacevic, N., Cohn, M., & McAndrews, M. P. (2013). Characterizing Functional Integrity: Intraindividual Brain Signal Variability Predicts Memory Performance in Patients with Medial Temporal Lobe Epilepsy. *Journal of Neuroscience*, 33(23), 9855-9865. <https://doi.org/10.1523/Jneurosci.3009-12.2013>
- Rakic, P. (2003). Developmental and evolutionary adaptations of cortical radial glia. *Cereb Cortex*, 13(6), 541-549. <https://doi.org/10.1093/cercor/13.6.541>
- Rieck, J. R., DeSouza, B., Baracchini, G., & Grady, C. L. (2022). Reduced modulation of BOLD variability as a function of cognitive load in healthy aging. *Neurobiology of Aging*, 112, 215-230. <https://doi.org/10.1016/j.neurobiolaging.2022.01.010>
- Roberts R.P., W. K., Moreau D., Addis D.R. (2023). Reassessing the Functional Significance of BOLD Variability. *bioRxiv*. <https://doi.org/doi:https://doi.org/10.1101/2023.02.06.527384>
- Sbairhat, H., Rajkumar, R., Ramkiran, S., Assi, A. A., Felder, J., Shah, N. J., Veselinovic, T., & Neuner, I. (2022). Test-retest stability of spontaneous brain activity and functional connectivity in the core resting-state networks assessed with ultrahigh field 7-Tesla resting-state functional magnetic resonance imaging. *Hum Brain Mapp*, 43(6), 2026-2040. <https://doi.org/10.1002/hbm.25771>
- Scarapicchia, V., Mazerolle, E. L., Fisk, J. D., Ritchie, L. J., & Gawryluk, J. R. (2018). Resting State BOLD Variability in Alzheimer's Disease: A Marker of Cognitive Decline or Cerebrovascular Status? *Frontiers in Aging Neuroscience*, 10, 39. <https://doi.org/10.3389/fnagi.2018.00039>
- Uddin, L. Q. (2020). Bring the Noise: Reconceptualizing Spontaneous Neural Activity. *Trends Cogn Sci*, 24(9), 734-746. <https://doi.org/10.1016/j.tics.2020.06.003>
- Vakorin, V. A., Lippe, S., & McIntosh, A. R. (2011). Variability of brain signals processed locally transforms into higher connectivity with brain development. *J Neurosci*, 31(17), 6405-6413. <https://doi.org/10.1523/JNEUROSCI.3153-10.2011>
- Wang, H., Ghaderi, A., Long, X., Reynolds, J. E., Lebel, C., & Protzner, A. B. (2021). The longitudinal relationship between BOLD signal variability changes and white matter maturation during early childhood. *NeuroImage*, 242, 118448. <https://doi.org/10.1016/j.neuroimage.2021.118448>
- Whittaker, J. R., Driver, I. D., Bright, M. G., & Murphy, K. (2016). The absolute CBF response to activation is preserved during elevated perfusion: Implications for neurovascular coupling measures. *NeuroImage*, 125, 198-207. <https://doi.org/10.1016/j.neuroimage.2015.10.023>
- Wu, X., Wei, L., Wang, N., Hu, Z., Wang, L., Ma, J., Feng, S., Cai, Y., Song, X., & Shi, Y. (2016). Frequency of Spontaneous BOLD Signal Differences between Moderate and Late Preterm Newborns and Term Newborns. *Neurotox Res*, 30(3), 539-551. <https://doi.org/10.1007/s12640-016-9642-4>

- Zang, Y. F., He, Y., Zhu, C. Z., Cao, Q. J., Sui, M. Q., Liang, M., Tian, L. X., Jiang, T. Z., & Wang, Y. F. (2007). Altered baseline brain activity in children with ADHD revealed by resting-state functional MRI. *Brain Dev*, 29(2), 83-91. <https://doi.org/10.1016/j.braindev.2006.07.002>
- Zhang, C., Beste, C., Prochazkova, L., Wang, K., Speer, S. P. H., Smidts, A., Boksem, M. A. S., & Hommel, B. (2022). Resting-state BOLD signal variability is associated with individual differences in metacontrol. *Sci Rep*, 12(1), 18425. <https://doi.org/10.1038/s41598-022-21703-5>
- Zhao, N., Yuan, L. X., Jia, X. Z., Zhou, X. F., Deng, X. P., He, H. J., Zhong, J., Wang, J., & Zang, Y. F. (2018). Intra- and Inter-Scanner Reliability of Voxel-Wise Whole-Brain Analytic Metrics for Resting State fMRI. *Front Neuroinform*, 12, 54. <https://doi.org/10.3389/fninf.2018.00054>
- Zheng, W., Cui, B., Han, Y., Song, H., Li, K., He, Y., & Wang, Z. (2019). Disrupted Regional Cerebral Blood Flow, Functional Activity and Connectivity in Alzheimer's Disease: A Combined ASL Perfusion and Resting State fMRI Study. *Front Neurosci*, 13, 738. <https://doi.org/10.3389/fnins.2019.00738>
- Zou, Q. H., Zhu, C. Z., Yang, Y., Zuo, X. N., Long, X. Y., Cao, Q. J., Wang, Y. F., & Zang, Y. F. (2008). An improved approach to detection of amplitude of low-frequency fluctuation (ALFF) for resting-state fMRI: fractional ALFF. *J Neurosci Methods*, 172(1), 137-141. <https://doi.org/10.1016/j.jneumeth.2008.04.012>

REVIEWERS' COMMENTS

We thank sincerely the reviewers for their positive feedback. Detailed responses are provided below in blue.

Reviewer #1 (Remarks to the Author):

I thank the authors for carefully addressing my comments. I believe the updated version of this manuscript is much improved and will be of great interest to the field.

Reviewer #2 (Remarks to the Author):

It is well appreciated that the authors have done a great job in revising this manuscript. The authors have addressed previous comments by incorporating additional and relevant references on BOLD variability, conducting further analyses using other BOLD signal measures, providing more detailed methodological descriptions, and discussing the limitations related to the relatively limited dataset. The manuscript has been improved significantly after revision. Just a couple of minor comments remain as follows.

1. It would be helpful to clarify in the Methods section whether all features included in the LASSO regression were standardized prior to model fitting. Given the different scales of the diffusion-derived measures, explicit reporting of normalization is important. In addition, several of the microstructural metrics are likely highly correlated; therefore, the selection of MD as a predictor should be interpreted with caution. The authors may consider toning down language suggesting MD as the “best” predictor and instead emphasize that LASSO identifies representative features from correlated sets.

We thank the reviewer for their suggested, we have now clarified in the methods section that the features included in Lasso regression were normalized prior to model fitting. “For Lasso-based regression modelling, the same **normalized** group-level region \times metric matrix was used (**mean regional changes were first computed for each cortical resting-state network region and then normalized per metric by dividing by the largest absolute mean regional change**), but expressed as absolute values, in order to focus on the magnitude of change, irrespective of the direction (...).”

We also had already mention it in the following section: “The analysis was conducted on the group average 9 region \times 10 metric matrix ($n = 9$ regions), where BOLD variability change served as the dependent variable and nine microstructural metrics (extraMD, extrTrans, diff, intra, MD, FA, MK, AK, RK) served as predictors. **All metrics were standardized prior to modelling, as previously described.**”

We also agree with the reviewer suggestion regarding interpretability of Lasso results and we have removed the sentence: “(...) with mean diffusivity (MD) showing the most robust association with longitudinal changes in BOLD variability” from the discussion section. Now it can be read as follows. “Diffusion- and kurtosis-derived microstructural

metrics emerged as strong contributors. Under strict penalization, a limited set of predictors (MD, RK, AK and FA) was retained, explaining up to 28% of the observed changes in BOLD variability. Relaxing model sparsity increased the number of contributing microstructural metrics, including intra, diff, extraMD, MK, and substantially improved predictive performance up to 61–64%, suggesting that BOLD variability reflects a distributed and correlated microstructural maturation process. Together, these findings support an important role for cortical microstructural maturation in shaping functional variability.”

2. The Discussion cites prior studies applying NODDI to cortical gray matter. Since NODDI was established for white matter modeling and does not account for somas, making its application to cortical gray matter problematic. Previous studies tailored for cortical gray matter microstructure (e.g. PMID: 32289460; PMID: 41486548) have repeatedly found that NODDI might not be used appropriately for quantifying cortical microstructure. Considering these specific works, the authors should briefly acknowledge the methodological limitations and interpret NODDI-related findings with caution.

We thank the reviewer for their suggestion. We had already acknowledged the limitations of NODDI in cortical gray matter in the Introduction, when justifying our choice of SMT and DKI models. The paper cited in the Discussion primarily uses other microstructural models and reports that “primary sensory and motor cortices exhibited microstructural features indicating more advanced maturation, such as higher T1/T2 contrast, higher intracellular volume fraction, and lower mean diffusivity, compared with higher-order frontal and parietal regions.” These microstructural findings are broader than the individual NODDI intra-cellular volume fraction and retaining the reference in this context does not invalidate their results, despite the known limitations of NODDI. For this reason, we have kept this discussion point while already noting NODDI’s limitations in the Introduction.

Reviewer #3 (Remarks to the Author):

The authors have addressed my questions.